# HLA-class II restricted TCR targeting human papillomavirus type 18 E7 induces solid tumor remission in mice

Jianting Long[1,4], Xihe Chen[2,4], Mian He[3], Shudan Ou[2], Yunhe Zhao[3], Qingjia Yan[2], Minjun Ma[2], Jingyu Chen[1], Xuping Qin[1], Xiangjun Zhou[2], Junjun Chu ®[2] ✉ & Yanyan Han ®[2] ✉

T cell receptor (TCR)-engineered T cell therapy is a promising potential treatment for solid tumors, with preliminary efficacy demonstrated in clinical trials. However, obtaining clinically effective TCR molecules remains a major challenge. We have developed a strategy for cloning tumor-specific TCRs from long-term surviving patients who have responded to immunotherapy. Here, we report the identification of a TCR (10F04), which is human leukocyte antigen (HLA)-DRA/DRB1*09:01 restricted and human papillomavirus type 18 (HPV18) E7$_{84-98}$ specific, from a multiple antigens stimulating cellular therapy (MASCT) benefited metastatic cervical cancer patient. Upon transduction into human T cells, the 10F04 TCR demonstrated robust antitumor activity in both in vitro and in vivo models. Notably, the TCR effectively redirected both CD4$^+$ and CD8$^+$ T cells to specifically recognize tumor cells and induced multiple cytokine secretion along with durable antitumor activity and outstanding safety profiles. As a result, this TCR is currently being investigated in a phase I clinical trial for treating HPV18-positive cancers. This study provides an approach for developing safe and effective TCR-T therapies, while underscoring the potential of HLA class II-restricted TCR-T therapy as a cancer treatment.

Cervical cancer is a significant global public health concern, ranking as the second most common malignancy affecting women's reproductive organs after uterine cancer. It is attributed to high-risk subtypes of human papillomavirus (HPV) infection with more than 95% of cases[1]. In 2020, there were ~604,000 new cases of cervical cancer, with 342,000 deaths reported worldwide, and 90% of these cases occurred in low- and middle-income countries[2]. Even in high-income countries, the 5-year survival rate for distant metastatic cervical cancer remains less than 20%, and there has been no significant improvement since the 1970s due to the lack of effective treatment options[3]. Therefore, there is an urgent need to develop therapeutic approaches for late-stage cervical cancer.

Immunotherapy, which utilizes the patient's immune system to recognize and attack cancer cells, has emerged as a rapidly advancing technology in cancer treatment over the past decades. Immune checkpoint inhibitors (ICIs), which block the immunosuppressive CTLA-4 or PD1 signal pathway of T cells, have been approved and widely used for various types of cancers[4], including cervical cancer[5]. While some patients experience remarkable remissions, the overall response rate (ORR) of anti-PD1 monotherapy for PDL1-positive cervical cancer is only 14.3%[5]. One of the main reasons limiting the therapeutic efficiency of ICIs is the lack of pre-existing T cells capable of recognizing and eliminating cancer cells[6]. Vaccination, as a safe and

[1]Department of Oncology, Cancer Center, The First Affiliated Hospital, Sun Yat-sen University, Guangzhou, PR China. [2]HRYZ Biotech Co., Guangzhou, PR China. [3]Department of Obstetrics and Gynecology, The First Affiliated Hospital, Sun Yat-sen University, Guangzhou, PR China. [4]These authors contributed equally: Jianting Long, Xihe Chen. ✉e-mail: chujunjun@shhryz.com; hanyanyan@shhryz.com

reliable method, can induce specific T cells not only to prevent HPV infection and HPV-associated cancer[7] but also to target existing tumors[8,9]. The effectiveness of different types of therapeutic cancer vaccines, such as the dendritic cell (DC) vaccine, mRNA vaccine, and peptide vaccine, has been demonstrated in clinical trials for multiple malignancies[9].

Another approach to obtain tumor-specific T cells is through the adoptive transfer of in vitro induced or genetically engineered T cells. One of the most successful attempts is the use of chimeric antigen receptor (CAR)-T cells, with six CAR-T therapies having been approved by the Food and Drug Administration (FDA) since 2017[10]. While CAR-T cells have shown success in hematologic malignancies, their efficacy remains limited and significant risks may exist in solid tumors. In comparison, T cell receptor (TCR)-engineered T cells have a broader range of target antigens and are considered more promising against solid tumors[11]. The first TCR-T cell therapy (Afami-cel) targeting MAGE-A4 to treat synovial sarcoma has shown impressive therapeutic effects in phase I/II clinical trials and is currently being submitted for biologics license application (BLA) to FDA[12]. Furthermore, TCR-T cells targeting NY-ESO-1 achieved an average response rate of 47% in a large cohort containing 107 melanoma and synovial sarcoma patients[11]. A similar result was reported in a phase I trial of NY-ESO-1 TCR-T cell therapy in advanced soft tissue sarcoma[13]. Although tumor-reactive CD4 T cells have been identified for decades[14], almost all of the TCR-T therapy developed in clinical stage are HLA-I restricted CD8 TCRs. CD4 T cells used to be considered helper T cells that assist other immune cells in eliminating tumors. However, accumulating evidence from clinical trials has shown that HLA-II restricted TCR transduced TCR-T cells or CD4 tumor-infiltrating lymphocytes (TILs) are promising treatment for malignant cancers[15,16].

Multiple antigens stimulating cellular therapy (MASCT) is an immune therapeutic approach that combines dendritic cells (DCs) and autologous T cells. Autologous monocytes from patients were induced to immature dendritic cells using GM-CSF and IL4, and subsequently pulsed with ~15 tumor antigen peptides. These DCs were then matured through stimulation with Poly I:C or MPLA and cytokines. The antigen-loaded mature DCs were used for direct injection or continued in vitro induction of tumor-specific T cells. The expanded tumor-specific T cells were then infused. Cancer patients typically undergo multiple cycles of MASCT treatment, each consisting of 3 sequential infusions of DCs and 3 infusions of T cells. MASCT is currently being investigated in phase II clinical trials. In our previous clinical studies, we have observed dynamic and specific T cell immune responses against certain tumor antigens in patients with hepatocellular carcinoma (HCC) after treatment with MASCT[17]. Furthermore, MASCT treatment has been reported to significantly improve the overall survival of HCC patients[18]. One of the major challenges in TCR-T therapy is identifying TCRs with high affinity and specificity, however, natural TCR with high affinity is rare. Long-term surviving patients are considered a valuable source for identifying natural therapeutic TCRs with high potency[19]. Since MASCT-treated patients have received multiple rounds of tumor antigen stimulation, we hypothesize that it is highly possible to clone therapeutic TCRs from patients who have benefited from MASCT and have shown a good immune response through continuous immune surveillance.

To this end, we have successfully identified tumor antigen-specific TCRs from MASCT-benefited cancer patients. Here, we report a TCR (10F04) identified from a HPV18-positive patient with metastatic cervical cancer who has shown a strong immune response and long-term survival. This TCR specifically recognizes HLA-DRA/DRB1*09:01 restricted HPV18 E7$_{84-98}$ peptide. Robust and durable antitumor activity was observed both in cell-based assays and mouse xenograft models when the 10F04 TCR was transduced to human T cells. Importantly, no cross-activity against other HLA and human genes was identified. Herein, we concluded that 10F04 TCR-T is a highly promising therapy against HPV18 E7-positive cancers.

## Results

### The sustained immune responses of a long-term survived patient with metastatic cervical cancer after MASCT treatment

A patient was diagnosed with human papillomavirus (HPV) positive metastatic cervical squamous cell carcinoma in 2011. The patient underwent radical resection followed by adjuvant chemo-radiation therapy. However, 33 months after the initial diagnosis, metastasis was detected on the right sacroiliac joint. As standard therapy failed, the patient had repeatedly received MASCT treatments since 2014 (Fig. 1a). Fifteen tumor-associated antigens, along with peptides from three HPV subtypes, were utilized for dendritic cell loading and induction of specific T cells. The metastatic bone tumor was effectively suppressed after MASCT treatments resulting in sustained tumor-free survival for over 9 years (Fig. 1a, b). To evaluate the immune response during and after MASCT treatment, the patient's peripheral blood mononuclear cells (PBMCs) were continuously collected for immune surveillance using IFNγ ELISPOT assay. Significant and durable T cell immune responses were observed in the patient's peripheral blood at different time points against diverse tumor antigens, including the p53, carcinoembryonic antigen (CEA), regulator of G-protein signaling 5 (RGS5), and HPV18E7 (Supplementary Fig. 1). In particular, a strong and persistent T-cell response against multiple HPV antigens was observed after MASCT treatment, especially for the HPV18E7$_{76-105}$ peptide (Fig. 1c, d).

Taken together, a robust and sustained immune response has been observed during and after MASCT treatment. This suggests that those existing T cells that recognized HPV18E7 antigen might contribute to the long-term survival of this late-stage cervical cancer patient.

### Identification of an HPV18 antigen-specific TCR

To identify potential therapeutic TCRs, we developed a Reverse Genetic Engineering of the TCR-T (ReGET) platform. Briefly, PBMCs from this patient were collected and stimulated with autologous dendritic cells pulsed with HPV18E7$_{76-105}$ peptide in vitro. After several rounds of stimulation, T cells specific against HPV18E7$_{76-105}$ peptide were sorted out for bulk and single-cell TCR sequencing (scTCRseq). Eventually, 56 α and β chain cognate paired TCRs were cloned, and 5 HPV18E7$_{76-105}$ peptide recognizing TCRs were identified through functional assay experiments. When stimulated with HPV18E7$_{76-105}$ peptide-loaded autologous lymphoblastoid cell lines (LCL), 10F04 and P09B08 TCR transduced T cells secrete multiple cytokines (TNFα and IFNγ) (Supplementary Fig. 2a, b).

HPV18E7$_{76-105}$ is a long peptide containing 30 amino acids. The exact epitope recognized by 10F04 and P09B08 TCRs was identified through analysis of serial truncated peptides, and ultimately determined to be HPV18E7$_{84-98}$ (Fig. 2a and Supplementary Fig. 2c). HLA blocking experiment revealed that inhibiting HLA-DR molecules in LCLs led to a significant reduction in IFNγ secretion by 10F04 and P09B08 TCR-T cells (Fig. 2b and Supplementary Fig. 2d). The HLA restriction of 10F04 and P09B08 TCRs was further investigated by individually overexpressing all class II HLA molecules of this patient in HEK-293T cells. Results showed that both TCRs recognize the HLA-DRA/DRB1*09:01-presented HPV18E7$_{84-98}$ epitope (Fig. 2c and Supplementary Fig. 2e). Additionally, peptide titration assays revealed a higher functional avidity of 10F04 TCR transduced T cells, which has been selected for further study (Fig. 2d and Supplementary Fig. 2f).

HeLa is an HPV18-positive cervical carcinoma cell line and endogenously expresses the HPV18 E7 protein[20]. To investigate if 10F04 could recognize endogenously expressed HPV18E7 protein other than HPV18E7$_{84-98}$ peptide, HeLa-DR0901 cells and HeLa cells overexpressing HPV18E7 protein (HeLa-DR0901-HPV18E7) were used to

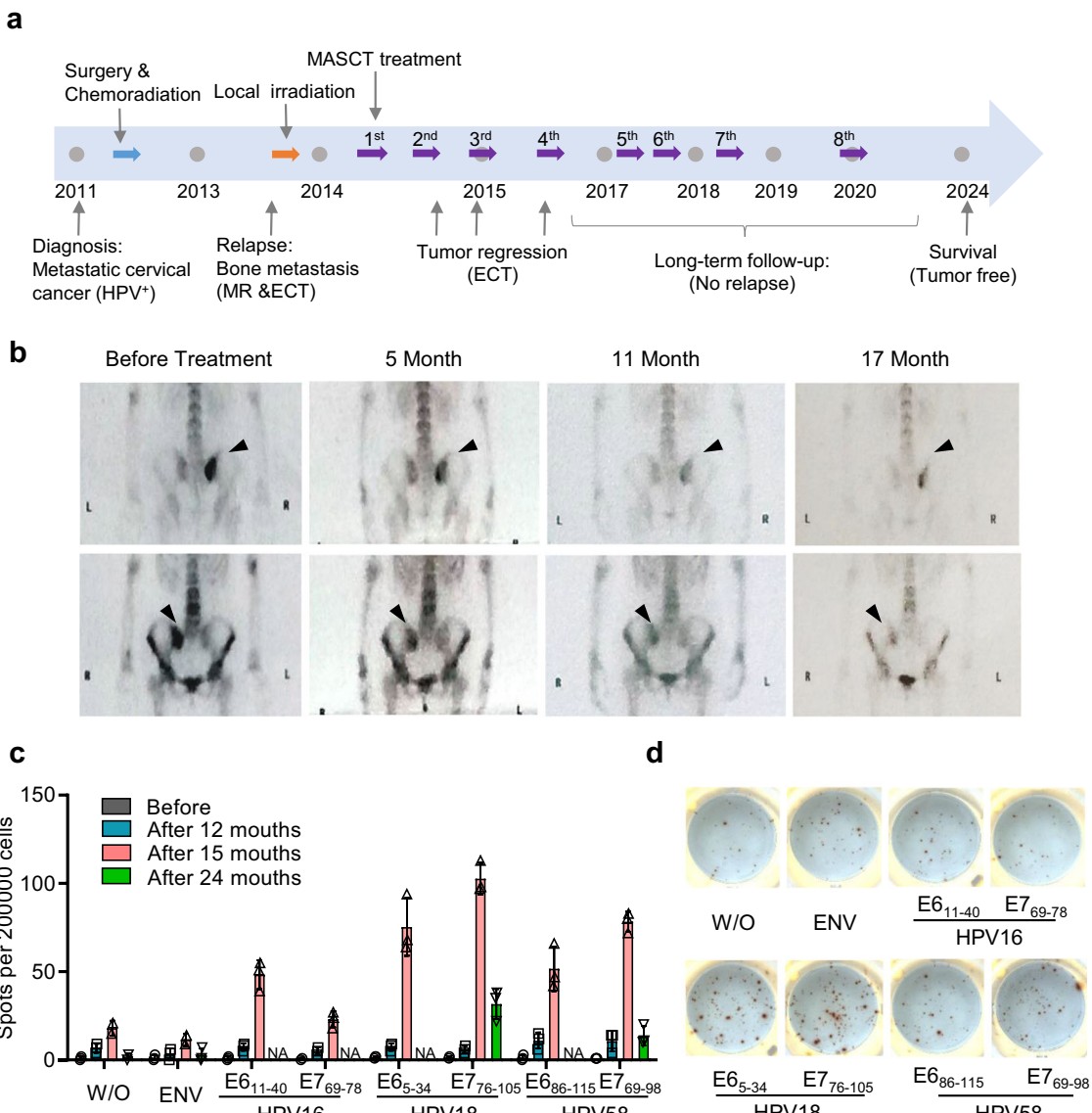

**Fig. 1 | Durable T cell immune response against HPV derived antigens of a long-term survived metastatic cervical cancer patient after MASCT treatment. a** The schematic diagram of the therapy history for a patient with metastatic cervical cancer who had previously failed standard treatment and presented with bone metastasis on the right sacroiliac joint. After local irradiation, the patient repeatedly received eight courses of MASCT treatments in the following 6 years. Each course of treatment contains three injections for multiple antigen peptide-pulsed autologous mature DCs and three injections for peptide-pulsed mature DCs-induced autologous T cells. **b** Emission computed tomography (ECT) scanning images of the cervical cancer patient before and after MASCT treatment. **c** Ex vivo screening of immune responses against HPV antigen peptides of the HPV-positive cervical cancer patient treated with MASCT immunotherapy. IFNγ−ELISPOT was performed to detect specific immune responses against HPV16/18/58 peptides of the patient's PBMCs. W/O without peptide, ENV irrelevant peptide control, NA not available. **d** Representative images of IFNγ ELISpot wells of the HPV18 E6/E7 immune responses at 15 months after MASCT immunotherapy. Data are shown as the mean ± SD, $n$ = 3 technical replicates (**c**). Source data are provided as a Source Data file.

stimulate 10F04 TCR-T cells (Supplementary Fig. 3a, b). We found that 10F04 transduced T cells are able to recognize endogenously processed HPV18E7 epitope presented on HLA-DRA/DRB1*09:01 (Fig. 2e). Moreover, over-expression of HPV18E7 protein further increased the target-specific recognition (Fig. 2e).

In summary, we demonstrated that 10F04 is a HLA-DRA/DRB1*09:01 restricted TCR capable of recognising both synthesized HPV18E7$_{84-98}$ peptides and HPV18E7$_{84-98}$ epitope naturally processed from endogenously expressed HPV18E7 protein in tumor cells.

### 10F04mc TCR-T displays superior antitumor potency in vitro and in vivo

The 10F04 TCR can be expressed on both CD4$^+$ and CD8$^+$ human T cells, but its recognition activity against tumor cells is still not strong enough (Supplementary Fig. 2a, b), which probably due to the inefficient pairing of transduced 10F04 TCR of the surface of T cells. It has been reported that murinization of human TCRs, by replacing human constant regions with murine ones, can avoid mispairing with endogenous TCR and improve TCR function[21]. Thus, the murinization and codon optimization of 10F04 TCR were performed. Indeed, we found that the surface expression of TCR has increased (Fig. 3a and Supplementary Fig. 2a), as well as the target cell recognition activity has been dramatically enhanced (Fig. 3b and Supplementary Fig. 2b). When co-cultured with target cells, optimized TCR (named 10F04mc TCR) transduced T cells produced high-level IL2, TNFα, and Granzyme B as well indicating a poly-functional profile of 10F04mc TCR-T cells (Fig. 3c). To investigate the tumor-killing capability of 10F04mc TCR-T cells, a cytotoxicity assay was performed using a Real-Time Cell

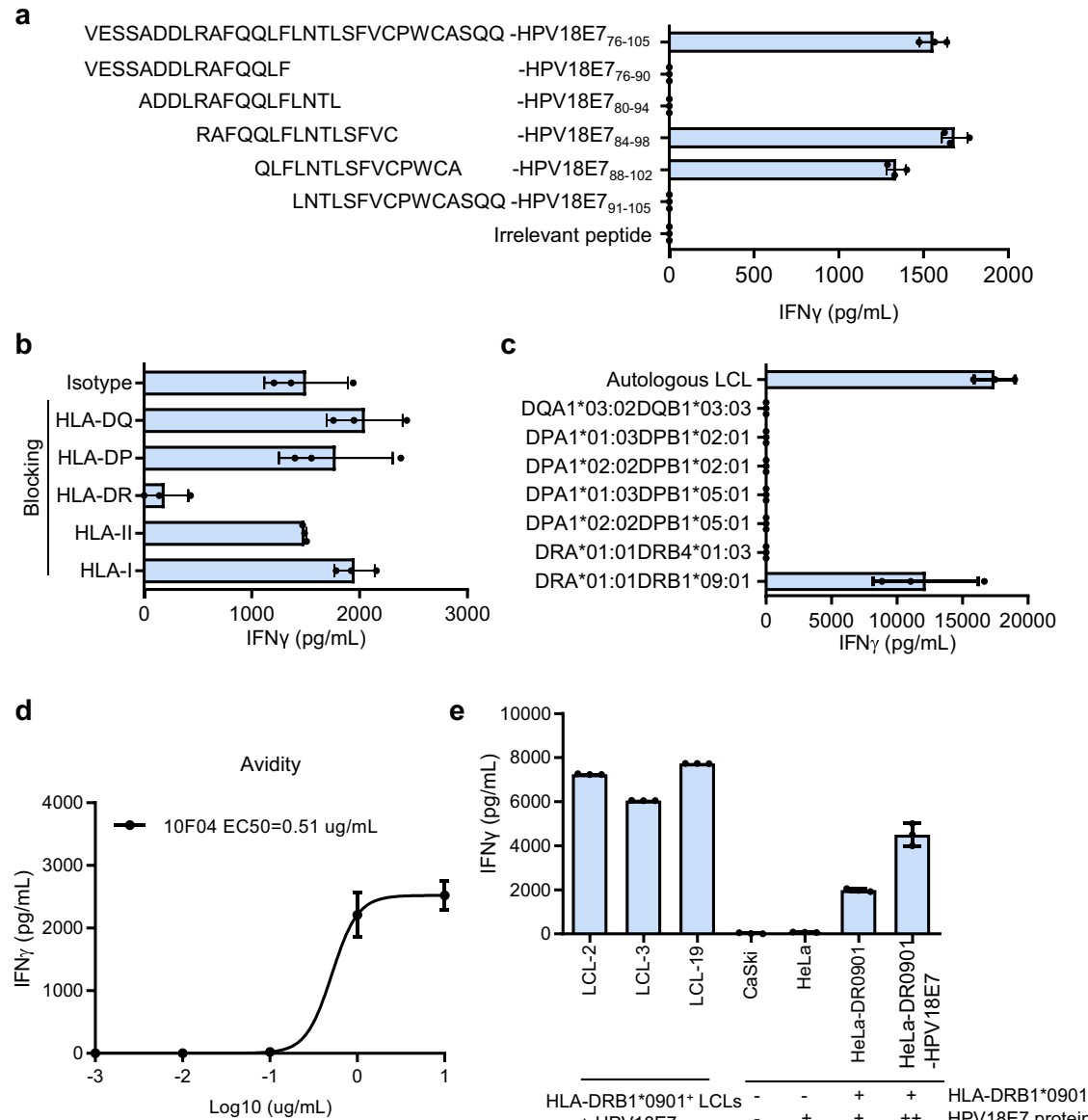

**Fig. 2 | Identification of an HPV18E7$_{84-98}$-specific TCR restricted to HLA-DRA/DRB1*09:01. a** Epitope identification of 10F04 TCR. Series truncated peptides derived from HPV18E7$_{76-105}$ were pulsed on autologous LCLs to stimulate 10F04 transduced T cells. The concentration of IFNγ in the supernatant after overnight co-culture was detected by ELISA. **b** HLA restriction analysis of 10F04. The 10F04 transduced T cells were co-cultured with autologous LCLs pulsed with the HPV18E7$_{76-105}$ peptide in the presence of blocking antibodies or isotype control. The IFNγ secretion level was detected by ELISA. **c** 10F04 transduced T cells were co-cultured with HEK-293T cells expressing each HLA-II molecule of the autologous LCL, which were pulsed with HPV18E7$_{76-105}$ peptide. Autologous LCLs pulsed with

HPV18E7$_{76-105}$ peptide were used as a positive control. IFNγ secretion in the supernatant after overnight co-culture was detected by ELISA. **d** Avidity assay of 10F04 TCR transduced T cells. Serial diluted HPV18E7$_{76-105}$ peptide were used to assess the binding avidity of 10F04 and P09B08. IFNγ secretion was detected by ELISA. **e** Specific recognition analysis of 10F04 transduced T cells against HPV18E7⁺ cervical cancer cells. CaSki, and HeLa cell lines were incubated with 10F04 TCR-T cells and recognition was determined by the IFNγ ELISA analysis of the supernatant. HPV18E7$_{84-98}$ peptide pulsed HLA-DRA/DRB1*0901 positive LCLs were set up as positive control. Data are shown as the mean ± SD, $n = 3$ biologically independent samples (**a**–**e**). Source data are provided as a Source Data file.

Analyzer (RTCA) system at different effector and target (E: T) ratios. The result shows that 10F04mc TCR-T cells lysed HeLa-DR0901 cells in a dose-dependent pattern (Fig. 3d). The specific cytotoxicity against HPV18E7-positive tumor cells of 10F04mc TCR-T cells have been validated in multiple models of human cervical cancer (C-4I, SW756, MS751) and osteosarcoma cells U2OS (Supplementary Fig. 4a–i).

To assess the in vivo antitumor potential of 10F04mc TCR-T cells, severely immune deficient NOG (NOD.Cg-*Prkdc*$^{scid}$*IL2rg*$^{tm1Sug}$/ *JicCrl*) mice were engrafted with HeLa-DR0901-HPV18E7 cells administrated with a single intravenous (i.v.) injection of 10F04mc TCR-T. The results showed that tumor growth in mice treated with 10F04mc

TCR-T cells was significantly regressed (Fig. 3e). More importantly, 3 mice (1 of medium dose, 2 of high dose) are tumor-free, and another 3 mice (1 of medium dose, 2 of high dose) tumors are extremely small (tumor weight <0.1 g) at the end of the experiment (Supplementary Fig. 5a). Consistently, 10F04mc TCR-T cells demonstrated significant tumor-inhibition in the mouse xenograft experiments using endogenously HPV18-positive cell line models (Supplementary Fig. 3c–e).

Interestingly, we found a significant increase in the proportion of CD4⁺ TCR-T cells in the mice splenocytes at end of the experiment. Conversely, the proportion of CD8⁺ TCR-T cells showed a significant

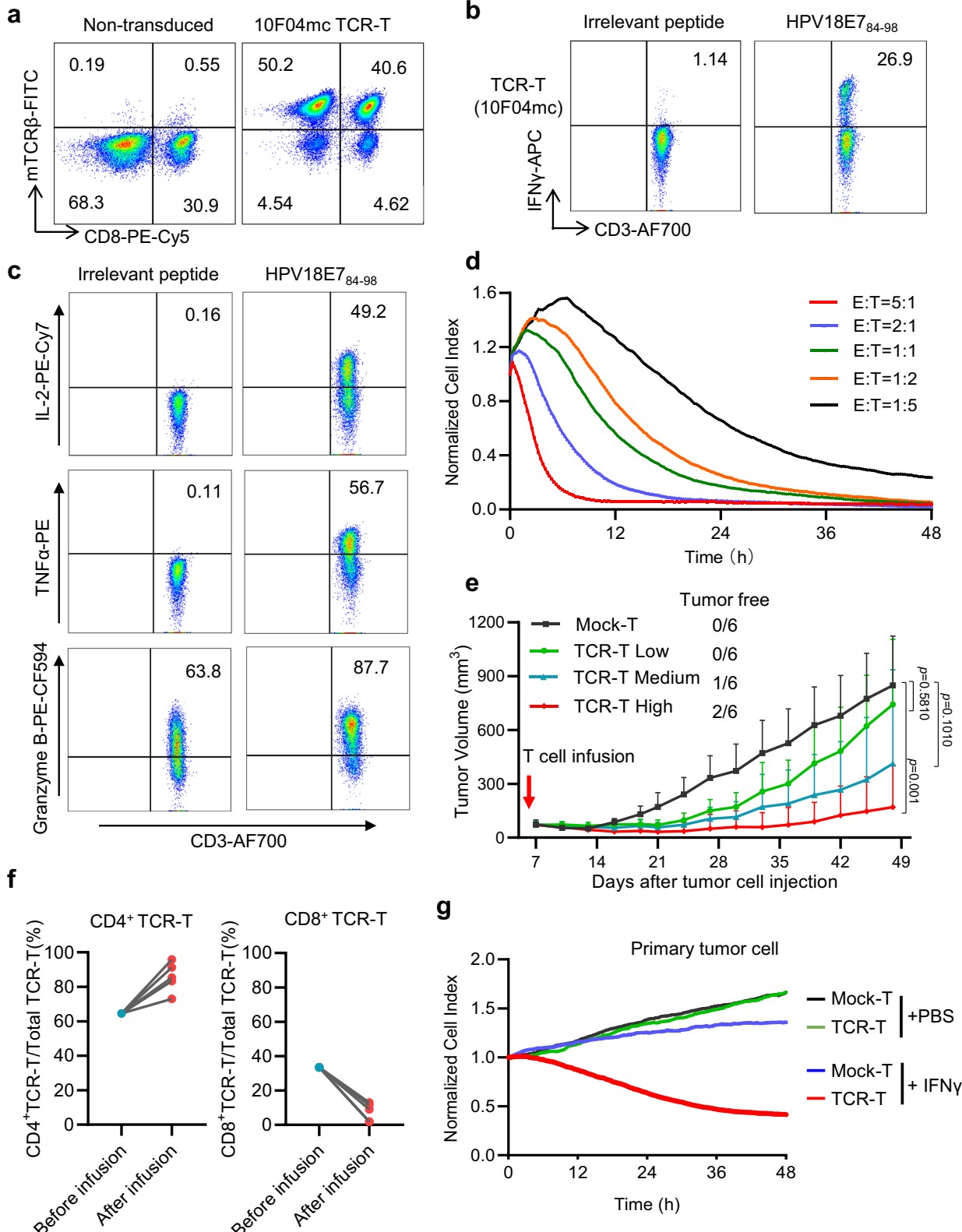

decrease (Fig. 3f). In the pharmacokinetic study in mouse model, we monitored the tissue distribution of injected 10F04mc TCR-T cells in the HeLa-DR0901-HPV18E7 tumor-bearing mice by absolute quantification PCR (qPCR). The result shows that 10F04mc TCR-T cells migrated to the tumor and expanded inside the tumor in the first 2 weeks. Interestingly, 10F04mc TCR-T cells were persistently present in the peripheral blood and eventually homed to peripheral immune organs (spleen and lymph nodes) for long-term survival (Supplementary Fig. 5b).

These findings suggest that 10F04mc TCR-T cells have potent antitumor activity in vivo and are capable of inducing tumor regression or complete tumor eradication in the murine model. Moreover, long-term survival of 10F04mc TCR-T cells may provide immune surveillance and prevent tumor recurrence.

**Fig. 3 | 10F04mc TCR-T displays superior antitumor potency in vitro and in vivo. a** Healthy donor-derived PBMCs were activated and transduced with constant region murinized and codon-optimized TCR (10F04mc). The TCR expression was assessed using flow cytometry with an anti-mouse TCRβ antibody. **b** K562-DRA/DRB1*09:01 cells were pulsed with HPV18E7$_{84-98}$ or irrelevant peptide and co-cultured with 10F04mc TCR-T cells. Intracellular IFNγ production of which was analyzed by flow cytometry assay. **c** Intracellular TNFα, IL-2, and Granzyme B production were analyzed similarly as described in (**b**). **d** The in vitro killing capability of 10F04mc transduced T cells was evaluated in different effector-to-target (E:T) ratios (from 1:5 to 5:1) by Real-Time Cell Analyzer, while HeLa-DR0901 cells were used as the target cells. **e** NOG mice were injected subcutaneously with $4 \times 10^6$ HeLa-DR0901-HPV18E7 cells per mouse at day 1. TCR-T groups received a single intravenous injection at a different dose (Low: $1 \times 10^7$, medium: $3 \times 10^7$, High: $5 \times 10^7$ 10F04mc TCR transduced T cells per mouse, respectively) 6 days after the tumor cells injection. Control group mice received $5 \times 10^7$ Mock-T cells at the same day.

The tumor volume was measured by digital caliper every 2–5 days. Mice were euthanized at day 48. Murine splenocytes of the high dose TCR-T treated mice (after infusion) and original TCR-T cells (before infusion) were analyzed by flow cytometry (**f**). **g** Primary cervical cancer cells were isolated from the surgical specimens of DRA/DRB1*0901$^+$cervical cancer patients and induced by IFNγ for 48 h in vitro. Next, primary cells were co-cultured with 10F04mc TCR-T cells (E:T = 2:1) for the cytotoxicity assay by the RTCA system. PBS treated primary cervical cancer cells and Mock-T cells were set up as negative controls. Data are representative of three independent experiments that resulted in similar results (**a–d**, **g**). Animal experiments were repeated twice under similar conditions with similar results (**e**, **f**). Data are shown as the mean + SD, $n = 6$ mice per group (**e**). For (**f**), $n = 5$ mice for after infusion groups. Statistical analysis was performed by two-tailed Student's $t$ test of the last measurement, with $p < 0.05$ considered significant (**e**). Source data are provided as a Source Data file.

## IFNγ treatment upregulates HLA-DR expression on tumor cells and sensitizes to 10F04mc TCR-T therapy

Recent studies have shown that although HLA-II molecules are considered to be less broadly expressed than HLA-I on nonimmune cells, there is significant expression of HLA-II on tumor cells[22]. Several studies have also linked HLA-II expression on tumors with increased antitumor immunity mediated by cytotoxic CD4 T cells[23,24]. Furthermore, it has been reported that antitumor therapy or IFNγ treatment can increase HLA-II expression[25,26]. Consistently, in this study, it was observed that HLA-DR expression was greatly enhanced in primary human cervical carcinoma cells after being treated with IFNγ for 48 h (Supplementary Fig. 5c). Meanwhile, the IFNγ-treated primary human cervical carcinoma cells showed increased sensitivity to 10F04mc TCR-T cells, as evidenced by enhanced tumor cell recognition and killing (Fig. 3g and Supplementary Fig. 5d). These findings suggest that IFNγ treatment can upregulate HLA-DR expression on tumor cells and enhance the recognition and cytotoxicity of 10F04mc TCR-T cells, supporting the potential of 10F04mc TCR-T cells for immunotherapy.

## Characterization of peptides and HLA cross-reactivity of 10F04mc TCR-T cells

Previous clinical studies of TCR-T therapies have reported undesirable toxicities caused by TCR cross-reactivity[27–30]. It is critical to evaluate the cross-reactivity of 10F04mc TCR transduced T cells, although 10F04 was directly identified from MASCT-benefit cancer patients without affinity maturation. Firstly, we identified the minimal core epitope of HPV18E7$_{88-98}$ (QLFLNTLSFVC). (Supplementary Fig. 6a). Next, alanine scanning was performed to evaluate the potential cross-reactivity of 10F04mc TCR-T cells. Alanine substitutions at positions 3, 4, 6, 8, and 9 resulted in more than 70% reduction of IFNγ secretion compared to stimulation with the core epitope, indicating the key motif is X-X-F-L-X-T-X-S-F-X-X (Fig. 4a). To further assess cross-reactivity, peptides of human proteins that shared residues with HPV18E7$_{88-98}$ at positions 3, 4, 6, 8, and 9 were identified through BLAST search. Six human peptides were synthesized and used for TCR-T recognition analysis, but no cross-reactivity against these peptides was detected (Fig. 4b). HLA cross-reactivity of the 10F04mc TCR-T was also evaluated by co-culturing 10F04mc TCR transduced T cells with LCLs expressing different HLA-I and HLA-II molecules pulsed with or without HPV18E7$_{88-98}$ peptide (Fig. 4c and Supplementary Table 1). Moreover, it was observed that 10F04 TCR-T cells only recognized HeLa cells transduced with HLA-DRA/DRB1*09:01 (HeLa-DR0901), but not any other HPV18-negative cancer cell lines, such as Huh7, A549, PANC1, HGC-27 and Lovo (Supplementary Fig. 6b, c). These findings suggest that 10F04mc TCR-T cells do not exhibit significant cross-reactivity with human peptides or HLA molecules.

More than 100 subtypes of HPV virus have been identified and many of which are linked to cervical cancer. Moreover, distinct HPV E7 proteins display sequence similarities. Therefore, we conducted a comprehensive comparison of HPV E7 protein sequences across all subtypes in the database and synthesized 11 peptides with the highest similarity to the HPV18 E7$_{88-98}$ epitope (Supplementary Fig. 7a). Through peptide recognition experiments, we found that the 10F04mc TCR-T cells also recognize the HPV45 E7$_{89-99}$ epitope (QLFLSTLSFVC), whereas the recognition of the HPV85 E7$_{91-101}$ epitope (QLFLGTLSFLC) is notably weak (Supplementary Fig. 7b). Despite sharing the same core epitope with HPV18 E7$_{88-98}$, HPV45 E7$_{89-99}$, and HPV85 E7$_{91-101}$ exhibit 1 or 2 different amino acids. The avidity of 10F04mc TCR-T against HPV45 E7$_{89-99}$ is slightly weaker compared to the HPV18 E7$_{88-98}$ epitope (Supplementary Fig. 7c).

More importantly, we analyzed the existence of the 10F04 TCR sequence at different time points in PBMC samples of the patient through TCR-sequencing. The result shows that the 10F04 TCR sequence was continuously detected (Table 1). The long-term persistence of the endogenous 10F04 TCR expression T cells indicated that this naturally existing TCR was well tolerated in human. To expand the use of 10F04 TCR-T therapy for other HPV18-positive cancer patients, the safety and efficacy will be further investigated in the following clinical trials.

## 10F04mc TCR induces multifunctional and durable antitumor activity in CD4 T cells

We have confirmed that 10F04mc TCR is class II HLA molecular restricted but can be expressed on the surface of both CD4$^+$ and CD8$^+$ human T cells (Fig. 3a). When co-cultured with target cells, 10F04mc TCR-T cells secret cytokines and directly induce target cell lysis (Supplementary Movie 1 and 2). Thus, we sought to check if 10F04mc TCR is functional in both CD4$^+$ and CD8$^+$ human T cells. Transduced T cells were divided into CD4$^+$ TCR-T and CD8$^+$ TCR-T populations and their functions were studied, respectively. The results of flow cytometry analysis showed that CD4$^+$ TCR-T cells expressed higher levels of TNFα, IL-2, and Granzyme B but lower levels of IFNγ compared to CD8$^+$ TCR-T cells when stimulated by the specific antigen (Fig. 5a and Supplementary Fig. 8). Further, ELISA assays confirmed that CD4$^+$ TCR-T cells secreted significantly more IL-2 compared to CD8$^+$ TCR-T cells, which was consistent with the results of cytokine expression tested by flow cytometry (Supplementary Fig. 9a). However, the pattern of IFNγ secretion was quite the opposite. IFNγ secretion of CD4$^+$ TCR-T cells was significantly more than that of CD8$^+$ TCR-T cells (Supplementary Fig. 9a), which may due to the rapid response of CD8$^+$ TCR-T cells and the continuous response of CD4$^+$ TCR-T cells. Meanwhile, more memory phenotype (Tem) TCR-T cells were found in the CD4$^+$ population compared to the CD8$^+$ population after being stimulated with target tumor cells (Fig. 5b).

To further investigate the tumor-killing capability of CD4$^+$ TCR-T and CD8$^+$ TCR-T T cells, a repeating challenge experiment was performed (4 rounds and 36 h per round). After each round, the target cells were replaced with fresh HeLa-DR0901 cells, while the effector

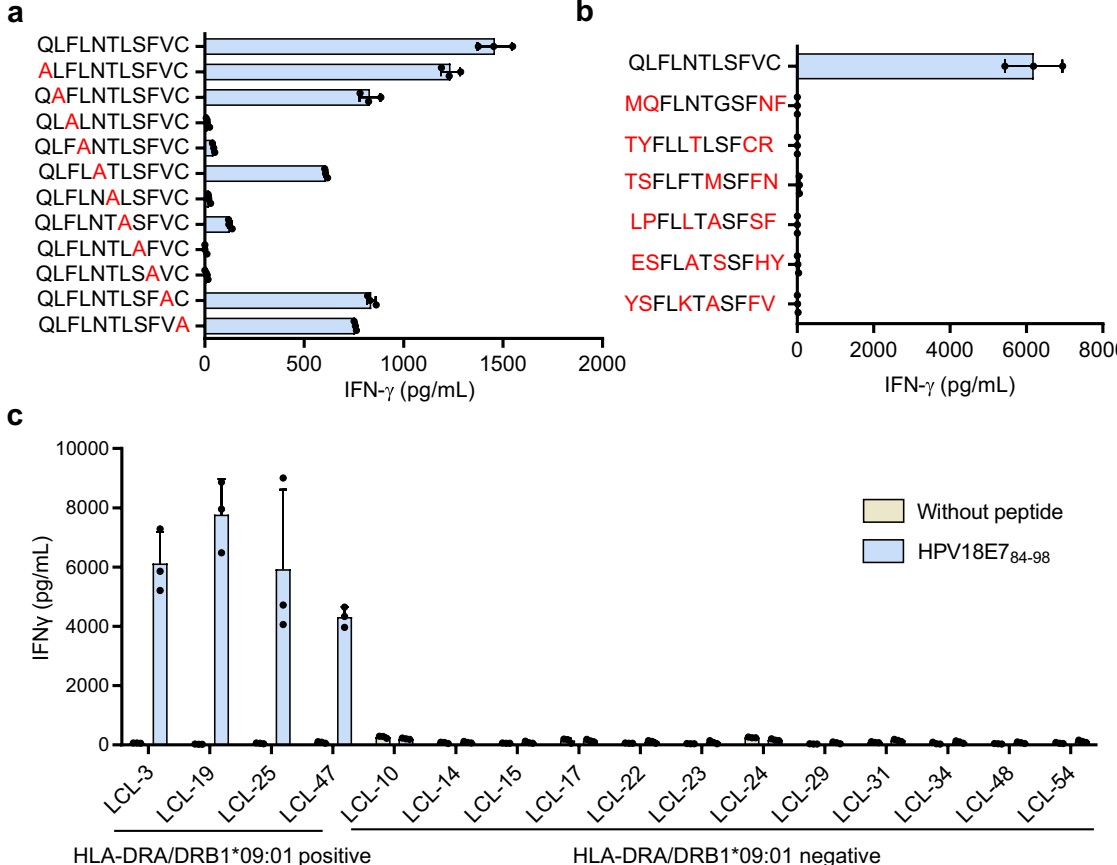

**Fig. 4 | No cross-reactivity of 10F04mc TCR-T cells. a** Alanine-scan screening to identify 10F04mc recognized key motif. A panel of synthetic peptides containing alanine substitution at all positions in the index HPV18E7$_{88-98}$ peptide were pulsed with K562-DR0901 cells. 10F04mc TCR-T cells were incubated with peptide pulsed K562-DR0901 cells for 16–24 h and IFNγ secretion was detected by ELISA. The alanine mutated position was considered as a key recognition site when the IFNγ secretion induced by this alanine mutated peptide was less than 30% of HPV18E7$_{88-98}$. The letters marked in red represent the mutated amino acids. **b** 10F04mc transduced T cells were tested by ELISA-IFNγ for recognition of 6 human peptides, which were identified by BLAST since their amino acids sequences were in accord

with the pattern of the key motif. The letters highlighted in red represent amino acids that are different from the key epitope. **c** HLA cross-reactivity analysis of 10F04mc TCR-T cells. 10F04mc transduced T cells were stimulated by LCLs with population representative HLA types pulsed with HPV18E7$_{88-98}$ peptide or without peptide. IFNγ secretion was detected by ELISA. HLA-DRA/DRB1*09:01 positive, $n = 4$ LCLs; HLA-DRA/DRB1*09:01 negative, $n = 12$ LCLs. The HLA typing information of all LCLs is shown in Supplementary Table 1. Data are shown as the mean ± SD (**a**–**c**), $n = 3$ biologically independent samples for different groups (**a**, **b**). Source data are provided as a Source Data file.

**Table 1 | Persistence of endogenous 10F04 TCR expression T cells in the MASCT-treated cervical cancer patient**

| Time | TCRα chains | | | TCR β chains | | |
|------|-------------|-------------------|------|--------------|-------------------|------|
|      | Total TCRα reads | 10F04 TCRα Reads | Rank | Total TCRβ reads | 10F04 TCRβ Reads | Rank |
| 2016 | 657,210 | 133 | 910/8054 | 914,309 | 253 | 652/16,134 |
| 2017 | 525,378 | 56 | 2373/7105 | 789,008 | 168 | 1168/6458 |
| 2018 | 1,403,890 | 439 | 709/8870 | 820,117 | 499 | 491/4019 |
| 2021 | 1,199,732 | 63 | 3975/22,771 | 782,426 | 29 | 6491/8054 |

cells remained the same. Interestingly, we found that 10F04mc TCR transduced CD4+ TCR-T cells alone were able to directly destroy target tumor cells (Fig. 5c). More importantly, multi-round experiments revealed that CD8+ TCR-T cells kill tumor cells faster in the 1st round, however, the tumor-killing capability of CD8+ TCR-T cells decreased progressively from the 2nd round. In contrast, CD4+ TCR-T cells displayed persistent and strong killing of target cells in the first 3 rounds and remained at 70% killing capability even in the 4th round (Fig. 5c and Supplementary Fig. 9b). In the course of the repeat challenge we observed that CD8 TCR-T cells exhibited rapid killing and expansion in the initial co-culture, followed by a decline in cell numbers and IFNγ secretion in subsequent rounds. In contrast, CD4 TCR-T cells

consistently demonstrated sustained killing efficacy, maintained high cell numbers, and consistently secreted IFNγ throughout the multiple rounds of co-culture (Fig. 5c, d and Supplementary Fig. 9b, c). Additionally, phenotype analysis found distinctive expression patterns of CD27 and CD39 in the two T cell subtypes, indicating potential differences in functionality (Supplementary Fig. 9d).

Consistently, a preclinical xenograft tumor model also demonstrated robust and durable in vivo antitumor activity of CD4+ TCR-T cells, whereas CD8+ TCR-T cells showed limited tumor regression compared to Mock T cells (Fig. 5e). Meanwhile, when we mixed the CD4+ TCR-T cells with CD8+ TCR-T cells at different ratios, and then repeatedly challenged with target tumor cells, we found that the

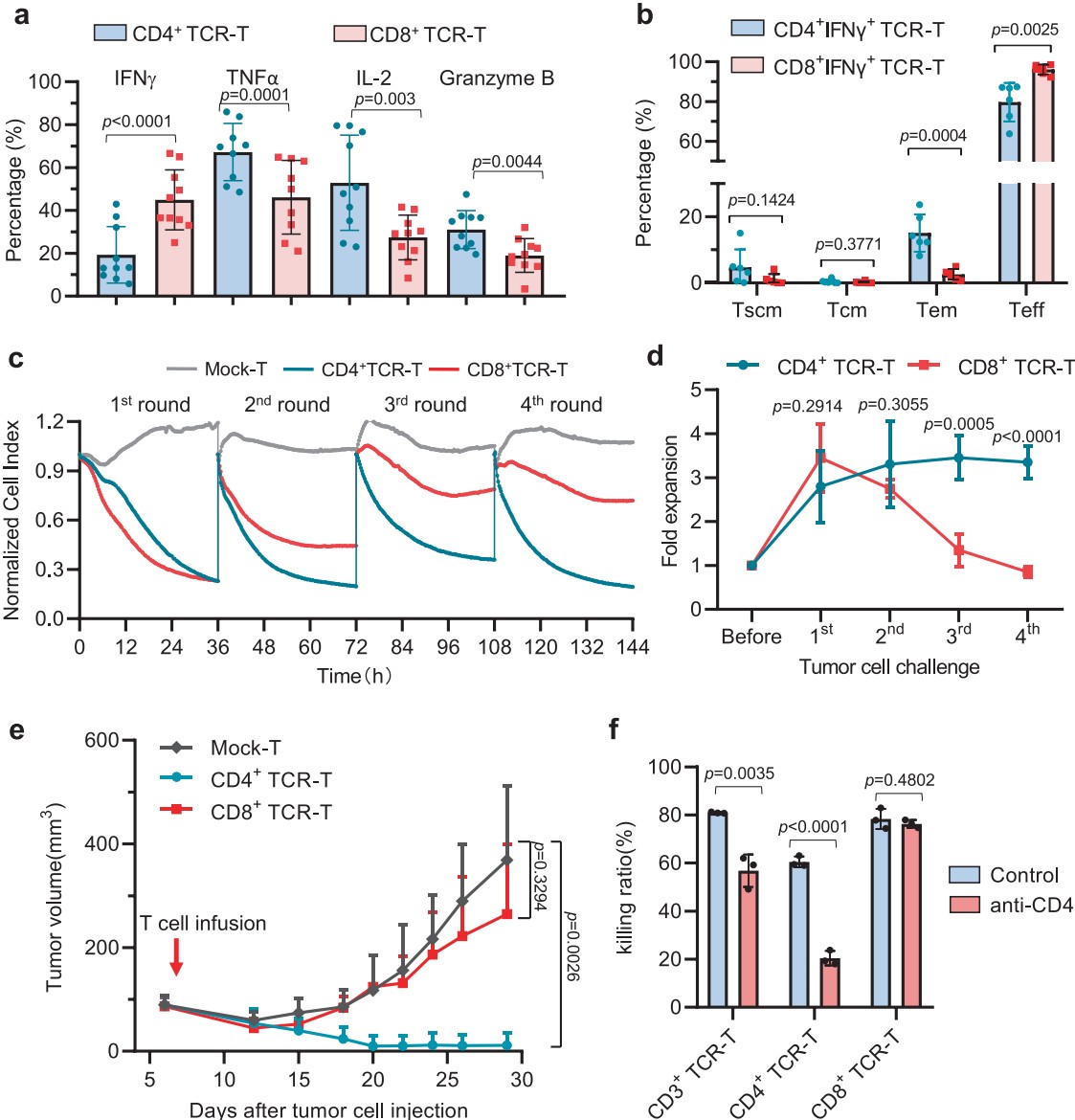

**Fig. 5 | 10F04mc TCR induces multifunctional and durable antitumor activity in CD4 T cells. a** 10F04mc transduced T cells were co-cultured with K562-DRA/DRB1*09:01 pulsed with HPV18E7$_{76-105}$ peptide or irrelevant peptide control, and the intracellular production IFNγ, TNFα, IL-2, and Granzyme B were analyzed by flow cytometry. Percentage number displayed is the HPV18E7$_{76-105}$ peptide with the irrelevant peptide subtracted. $N$ = 10 biologically independent samples for IFNγ, IL-2, and Granzyme B groups. $N$ = 9 biologically independent samples for TNFα groups. **b** The 10F04mc TCR-T cells were stimulated with K562-DRA/DRB1*09:01 pulsed with HPV18E7$_{76-105}$ peptide. Intracellular IFNγ and memory markers were stained and analyzed by flow cytometry. Tscm: stem-like memory T (CD45RO$^-$CCR7$^+$), Tcm: central memory T (CD45RO$^+$CCR7$^+$), Tem: effector memory T (CD45RO$^+$CCR7$^-$), Teff: effector T (CD45RO$^-$CCR7$^-$), $N$ = 6 biologically independent samples for different groups. **c** CD4$^+$/CD8$^+$ TCR-T cells were separated by positive selection after transduction and challenged by HeLa-DR0901 cells (E:T = 2:1). To evaluate the persistence of killing capability, the assay was continued

for 4 rounds (36 h per round). At the end of each round, T cells were transferred to a new RTCA plates well with the HeLa-DR0901 target cells. The representative cytotoxicity curves of four independent experiments that resulted in similar results were presented (**c**). **d** The proliferation of CD4$^+$ and CD8$^+$ TCR-T cells were counted after each round of challenge. $N$ = 4 biologically independent experiments. **e** A single intravenous injection (5 × 10$^7$ T cells per mouse) of CD3$^+$ TCR-T, CD4$^+$ TCR-T or CD8$^+$ TCR-T cells was applied 6 days after tumor cell injection. The tumor volume was measured every 3–5 days. Tumor growth curve were presented as means + SD, $n$ = 4 mice for each group. **f** Functional blocking of 10F04mc TCR-T cells with the anti-CD4 blocking antibody. The 10F04mc transduced CD4$^+$, CD8$^+$, or CD3$^+$ bulk T cells were challenged with HeLa-DR0901 cells in the presence of neutralizing CD4 antibody or isotype control. The target cell-killing ability was evaluated by the RTCA system. Data were presented as mean ± SD (**a**, **b**, **d**, **f**). Statistical significance was determined by two-tailed Student's $t$ test, with $p < 0.05$ considered significant (**a**, **b**, **d**, **e**). Source data are provided as a Source Data file.

percentage of CD4$^+$ TCR-T cells is positively correlated with robust and long-lasting anti-tumor efficacy both in vitro (Supplementary Fig. 10a) and in vivo models (Supplementary Fig. 10b). On the contrary, when we functionally block the CD4 molecule with an anti-CD4 antibody, the antitumor activity of both CD4$^+$ TCR-T cells and the bulk CD4$^+$ TCR-T cells dramatically decreases (Fig. 5f).

Based on these findings, we conclude that 10F04mc TCR is capable of redirecting both CD4 and CD8 T cells to specifically recognize target tumor cells. However, the persistent long-term antitumor activity appears to be mainly mediated by CD4+ TCR-T cells, as they demonstrated sustained cytokine secretion, tumor-killing capability, and memory phenotype compared to CD8$^+$ TCR-T cells.

## Discussion

HPV infection is responsible for causing 690,000 cases of cancer annually, accounting for 5% of all cancers worldwide[31]. The E6 and E7 proteins are viral oncoproteins encoded by the HPV genome, which are continuously expressed in infected cells[32]. These proteins play a critical role in the carcinogenesis of HPV-associated cancers by promoting cell cycle progression, inhibiting apoptosis, and interfering with cellular DNA repair mechanisms[33]. As a result, unlike other tumor-associated antigens, HPV E6/E7 proteins not only serve as biomarkers, but also drive cancer progression, making them attractive targets for therapeutic interventions. However, targeting E6/E7 proteins with traditional small molecule drugs poses challenges[33].

Viral proteins are usually highly immunogenic, making them good candidates for TCR therapy. In our long-term immune surveillance of MASCT clinical trial study, we observed a strong and durable anti-HPV-antigen-specific T cell response in the peripheral blood of HPV-associated cancer patients. This raises the possibility of cloning and utilizing these HPV antigen-specific T cells for treatment of other HPV-associated cancer patients. Both HPV E6 and E7 targeting TCR-engineered T cell therapy were evaluated in clinical trials against HPV-associated epithelial cancers[34,35]. In two comparative trials, HLA-A0201 restricted E7 TCR-T demonstrated a better objective clinical response (50%, 6/12) than E6 TCR-T (16.7%, 2/12), while both TCR-T approaches were found to be safe and well tolerated[34,35]. We also detected a stronger and more durable anti-E7 specific T cell response compared to anti-E6 antigen, suggesting that E7 antigen may be better targets for TCR based therapy.

CD4[+] T lymphocytes, in addition to CD8[+] T lymphocytes, have been found to play a significant role in tumor control and can be important in cancer immunotherapy[36–38]. CD4[+] T cells promote the proliferation, survival, and cytotoxicity of CD8[+] T cells by secreting different cytokines and on the other hand, sensitize tumor cells by enhancing the major histocompatibility complex (MHC) molecular expression and antigen presentation on tumor cells[39,40]. Moreover, CD4[+] T cells can acquire cytotoxic activity and exert tumoricidal functions by secreting cytotoxic proteins, such as Granzyme B, which is a critical effector protein of cytotoxic CD8[+] T and NK cells[41,42]. The 10F04mc TCR, which is a type II HLA molecular-restricted TCR, has been transduced into both CD4[+] and CD8[+] T cells, and both types of T cells have shown functional anti-tumor activity. CD4[+] T cells transduced with 10F04mc TCR not only secret cytokines such as IFNγ, TNFα, and IL-2 but also secret high levels of Granzyme B, indicating their potential to exert cytotoxic functions[41–43]. The multifunctional of 10F04 TCR in both CD4[+] and CD8[+] T cells may explain its superior antitumor activity.

Another advantage of CD4[+] T cell-based therapy is their longer persistence and long-term memory potential compared to CD8[+] T cells. Studies have shown that CD4[+] CAR-T cells can persist for decades in some patients with chronic lymphocytic leukemia who received CAR-T cell therapy[44,45]. CD8[+] T cells, on the other hand, are known to be fast killers but can quickly become exhausted after exposure to tumor cells. The 10F04mc CD4[+] TCR-T cells have been shown to exhibit persistent anti-tumor activity in long-term stimulation in both in vitro and in vivo models, and the TCRαβ chain sequence of 10F04 can be detected in peripheral blood of patients who received MASCT therapy for 7 years, indicating their durable anti-tumor ability. The durable antitumor ability for adoptive therapeutic T cells is more important when facing existing solid tumors compared with blood tumors. Therefore, the long-term in vivo persistence of 10F04 CD4[+] TCR-T makes it an attractive therapy for HPV18[+] malignant solid tumors.

TCR can recognize a wide range of targets of solid tumors and has shown remarkable responses in clinical studies[12,35]. The first TCR bispecific fusion protein targeting gp100 has been approved for the treatment of unresectable or metastatic uveal melanoma[46,47]. However,

concerns about the potential off-target toxicity of TCR therapy still exist[29,30]. The use of naturally occurring high-avidity TCRs, such as the 10F04 TCR identified in this study, from immunotherapy survivors may offer a safer alternative to artificially affinity-matured TCRs[19].

Most of the clinical stage TCR therapies are limited to HLA-A*02:01, which is more prevalent in Caucasian populations than in Asians. In contrast, HLA-DRB1*09:01, prevalent in the Asian population[48] (20.43%), particularly in Chinese population[49] (28.67%), presents an opportunity for broader use of the 10F04mc TCR therapies in this demographic. In addition to targeting the primary HPV18E7, 10F04m TCR-T also recognizes HPV45E7, which differs by a single amino acid. Given that ~5% of cervical cancer patients are HPV45-positive, making it one of the most prevalent HPV subtypes causing cervical cancer apart from HPV16 and HPV18 virus[50]. This suggests that the potential beneficiary population for 10F04mc TCR-T therapy may be further broadened.

The 10F04mc TCR, with its promising therapeutic potential and safety profile, is currently being tested in a phase I clinical study for the treatment of HPV18-positive cancers (NCT05952947). To the best of our knowledge, this is the first approved Investigational New Drug (IND) enabling HLA-II restricted TCR-T study in the file.

## Methods

### Ethical statement

This research complies with all relevant ethical guidelines and regulations. Human tissue samples and blood were collected following approval by the Ethics Committee (2014-01) of the First Affiliated Hospital, Sun Yat-sen University. All patients who provided samples gave written informed consent, agreeing to the use of personal clinical data and biological material for research purposes. All animal procedures were performed in accordance with relevant guidelines. Animal research protocols were approved by the Institutional Animal Care and Use Committee (IACUC) of Guangzhou Curegenix Inc (IACU-C#YSDW202203016-2).

### Primary cells and cell lines

EBV-LCLs were established by immortalization of B cells from PBMCs of patients and healthy donors using supernatant from the cell line B95.8. Briefly, $2 \times 10^5$ PBMCs cultured in RPMI-1640 medium (Gibco, cat. 11875093) supplemented with 10% FBS (Hyclone, cat. SH30084.03) were seeded in 96-well plates, and EBV supernatant was added in the presence of 1 ug/ml Cyclosporin A (Sigma, 1001274558) for 1 month.

Healthy donor derived PBMCs were purchased from Oribiotech (Shanghai, China). B95.8, HEK-293T, and HeLa cell lines were purchased from the Chinese Academy of Sciences. K562, C-4I, SW756, MS751, U2OS, LOVO, HGC27, A549 and PANC-1 cell lines were purchased from ATCC. HLA-DRB1*09:01 or HPV18E7 transduced cell line were established in house with lentivirus transduction and selected with 5 ug/ml puromycin (Biofeng Lab, cat. P8230-25mg) or 250 ug/ml G418 (Gibco, 10131-027). B95.8, K562, K562-DR0901, HGC27 and HGC27-DR0901 were cultured in RPMI-1640 medium supplemented with 10% FBS. HEK-293T, SW756, MS751, U2OS, Huh7, PANC-1 and HeLa cell lines were cultured in DMEM medium (Gibco, 11995065) supplemented with 10% FBS. C-4I cell lines were cultured in Waymouth's MB752/1 (Gibco, 11220-035) supplemented with 10% FBS. A549 and Lovo cell lines were cultured in F12K (Gibco, 21127022) supplemented with 10% FBS.

Primary cervical tumor cells were obtained from DRB1*09:01[+] patients. Fresh tumor tissue samples were enzymatically digested using a tumor dissociation kit (Gibco, cat. 17104019) according to the manufacturer's instructions, and tumor cells were cultured in RPMI-1640 medium supplemented with 10% FBS. All mediums contained 1% penicillin/streptomycin (Gibco, 15070063). Cells were maintained in a humidified atmosphere containing 5% $CO_2$ at 37 °C. For IFNγ

stimulation, primary human cervical carcinoma cells were treated with 4000 IU/mL recombinant Human IFN-γ (Peprotech, cat. 300-02) or PBS for 48 h.

### Antigen-specific T cells enrichment, bulk, and single-cell TCR sequencing

PBMCs from patients were stimulated with HPV18E7$_{76-105}$ peptide-pulsed autologous dendritic cells in vitro for 4 weeks. After stimulation, IFNγ$^+$ T cells were enriched by IFNγ Secretion Assay-Cell Enrichment and Detection Kit (Miltenyi, 130-054-201). One aliquot of IFNγ$^+$ T cells was performed single-cell TCR sequencing (scTCRseq) by iRepertoire. Another aliquot of IFNγ$^+$ T cells was performed Multiplex PCR for bulk TCR sequencing. Briefly, RNA extraction was carried out by RNA extraction reagent (QIAGEN, 74104) according to the manufacturer's instructions. CDR3 regions of α chain and β chain was amplified by using a bulk primer (iRepertoire, cat. HTAI-M-X-Ps, cat. HTBI-M-X-Ps) and One-step RT-PCR Kit (QIAGEN, cat. 210212) in the first round of PCR. PCR products were rescued by Ampure Beads (Beckman, cat. A63880) according to the manufacturer's instructions. The second round of PCR is carried out using communal primers (iRepertoire, cat. HTAI-M-X-Ps, cat. HTBI-M-X-Ps) and Multiplex PCR Kit (QIAGEN, cat. 206143).

Gel purified PCR product was sequenced by the Illumina MiSeq PE250 platform. Raw data were analyzed by iRepertoire pipeline (iRepertoire, Inc.) according to the data analysis guide (https://irepertoire.com/irweb-technical-notes/). The output data released by iRepertoire, Inc. consisted of the CDR3 amino acid sequence and the mapped V(D)JC genes and with reads counts. Combined CDR3 amino acid sequence and VJ genes (CDR3 a.a.-V-J) as a unique TCR clone then ranked them from highest to lowest by reads counts. PBMC samples from different time points of patients were also collected for bulk TCR sequencing as described above.

Peptides were chemically synthesized from BankPeptide Inc (Anhui, China) and Hybio Pharmaceutical (Shenzhen, China). Primers were synthesized from Sangon Biotech (Shanghai, China). Vectors were synthesized from Azenta (Suzhou, China).

### ELISPOT assay

The IFNγ ELISPOT assay was performed as previously described with some modifications[17,18]. Patients' PBMCs ($3 \times 10^5$ cells/well) were seeded in 96-well plates in AIM-V medium (Gibco, cat. A3021002) and stimulated with the antigen peptides or irrelevant peptides (10 μg/ml) for 48 h. Stimulated PBMCs were then transferred onto a 96-well ELISPOT assay plate (U-CyTech, cat. 230PR5) and stimulated again with peptides (10 μg/ml) for another 16–24 h for IFNγ detection. The ELISPOT assay was performed according to the manufacturer's instructions. The number of spot-forming units was detected by C.T.L. Immuno Spot S6 Analyzer and analyzed by Immuno Spot v6.0 software. The responses are represented as spot-forming units per $2 \times 10^5$ PBMCs/well.

### Lentiviral particle production

HEK-293T cells were seeded in a 10 cm tissue culture dish at a density of $5 \times 10^6$ cells per dish. The next day, HEK-293T cells were transfected with the TCR expressing plasmid and packaging plasmids (pMDLg/pRRE, pRSV-Rev, pMD2.G) using polyethylenimine (Polyplus, cat. PT-115-010). After 6–8 h of transfection, the medium was replaced with fresh DMEM supplemented with 10% FBS and 1% penicillin/streptomycin. After 48 h, the supernatant containing lentiviral particles was collected by centrifuging to remove cell debris, transferred to a fresh tube and filtered through a 0.45 μm filter to remove any remaining cell debris subsequently. Lentiviral particles were concentrated using ultracentrifugation at $20,000 \times g$ for 2 h at 4 °C, and resuspended in the culture medium. The titer of the lentiviral particles were determined using a lentiviral p24 ELISA kit (Takara, cat. 632200).

### T cell transductions

T cells were isolated from PBMCs using CD4-Microbeads (Miltenyi, cat. 130-045-101) and CD8-Microbeads (Miltenyi, cat. 130-045-201) at a ratio of 1:1. The isolated T cells were then activated using anti-CD3 (OKT3, Biolegend, cat. 317315) and anti-CD28 (CD28.2, Biolegend, cat. 302914) antibodies in the presence of 100 U/ml recombinant human interleukin-2 (IL-2; Peprotech, cat. 200-02-100ug) for 2 days. Activated T cells were transduced with the concentrated lentiviral particles at an MOI of 2–10. After 24 h, the cells were washed and cultured in complete T cell media (AIM-V supplemented with 10% FBS, 1% penicillin/streptomycin, and 1000 U/ml IL-2.) for further expansion. The transduction efficiency of T cells was determined by flow cytometry using a Hamster anti-mouse TCR βchain antibody (H57-597, BD Biosciences, cat. 553171) or anti-human TCR Vβ17 Antibody (E17.5F3.15.13, BECKMAN, cat. IM2048).

### Cytotoxicity assays

Cytotoxicity assays and target cell killing movies were performed using a Real-Time Cell Analyzer (RTCA) system. Briefly, RTCA plates (Agilent, cat. 300601020) were seeded with cancer cells at a density of 6000 cells/well. The plates were then placed in an RTCA analyzer (xCELLigence, ACEA Biosciences) and incubated at 37 °C with 5% $CO_2$ to allow for cell attachment and growth. After 16–24 h, TCR-T or Mock-T cells were added to the RTCA plates at a certain E:T ratio (Effector: Target). The plates were then placed in the RTCA analyzer and monitored for 36–48 h. For repetitive cytotoxicity assays, after 36 h of killing (round 1), cells in the well were collected and transferred to a new plate seeded with 6000 cells/well for another 36 h (round 2), so as to round 3 and round 4. The RTCA data were analyzed using the xCELLigence software (ACEA Biosciences). The killing kinetics were calculated by fitting the impedance data to a mathematical model that describes the interaction between T-cells and target cells.

### Surface and intracellular staining for flow cytometry

For surface staining, $1 \times 10^5$ - $1 \times 10^6$ T cells were collected and washed with FACS buffer (PBS containing 2% FBS). Surface staining antibodies were added and incubated at 4 °C for 30 min. After washing, samples were resuspended in FACS buffer for analysis by flow cytometer.

For intracellular cytokine staining, T cells were co-cultured with target cells at E:T ratio 1:1 for 4 h in the presence of GolgiPlug (BD Biosciences, cat. 00-4506-51). After co-cultured, surface markers were added to label cells for 30 min at 4 °C. These T cells were fixed, permeabilized in permeabilizing buffer (BD Biosciences, 554714) and stained with intracellular markers. Samples were acquired on BD FACS Canto II flow cytometer or Thermo Attune® NxT flow cytometer, and data were analyzed by using the Flowjo software.

The following antibodies were used for surface staining in this study: anti-human CD3-APC-Cy7 (SK7, BD Biosciences, cat. 557832), anti-human CD8-PercP (SK1, Biolegend, cat. 344708), anti-human CD8-PE-Cy5 (Hit8a, Biolegend, cat. 300910), anti-human CD4-PE-Cy7 (A161A1, Biolegend, cat. 357410), anti-human CD4-PE-Cy5 (RPA-T4, Biolegend, cat. 555348), CD3-Alexa Flour700 (SK7, Biolegend, cat. 344822), anti-mouse TCRβ chain-PE(H57-597, BD Biosciences, cat. 553172), anti-mouse TCRβ chain-FITC (H57-597, BD Biosciences, cat. 553171) and anti-human TCR Vβ17-PE (E17.5F3.15.13, BECKMAN, cat. IM2048), anti-human TCR Vβ2-PE (MPB2D5, BECKMAN, cat. IM2213), anti-human TCR Vβ1-PE (REA662, Miltenyi, cat. 130-110-018), anti-human HLA-DR-FITC (G46-6, BD, cat. 555811), anti-human CD27-BV421 (M-T271, Biolegend, cat. 356418), anti-human CD39-BV510 (A1, BD, cat. 567526), anti-human PD1-BV711 (MIH4, BD, cat.740814), anti-human CD3-BV570 (UCHT1, Biolegend, cat. 300436), anti-human CD4-ef450 (SK3, Invitrogen, cat. 48-0047-42), anti-human CD4-V500 (RPA-T4, BD, cat. 560768), anti-human CD8-AF700 (RPA-T8, BD, cat. 557945), anti-human CD8-APC-Cy7(RPA-T8, BD, cat. 557834), anti-mouse TCRvβ chain-APC-Cy7 (H57-597, BD, cat.560656).

The following antibodies were used for intracellular staining in this study: anti-human IFNγ-APC (B27, BD Biosciences, cat. 554702), anti-human TNFα-PE (MAb11, BD Biosciences, cat. 559321), anti-human Granzyme B-FITC (GB11, BD Biosciences, cat. 560211), anti-human Granzyme B-PE-CF594 (GB11, BD Biosciences, cat. 562062), anti-human IL-2-PE-Cy7 (MQ1-17H12, BD Biosciences, cat. 560707).

## Western blotting

Cells were lysed in RIPA lysis buffer (Beyotime, P0013B) and PMSF (Beyotime, ST506). Cell lysis was carried-out for 30 min on ice, lysates were then centrifuged to remove insoluble material. Then, the loading buffer (Beyotime, P0015) was added to the lysates and heated at 100 °C for 3–5 min to denature the protein completely. Samples were separated on 12% SDS-PAGE Gel (Beyotime, P0053A), transferred to PVDF membranes, and detected with the indicated antibodies. The following antibodies were used for western blotting: anti-HPV18 E7 antibody (8E2, abcam, ab100953, dilution: 1:1000), anti-β-actin antibody (2D4H5, proteintech, 66009-1-Ig, dilution: 1:5000), HRP-conjugated Affinipure Goat Anti-Mouse IgG(H + L) (proteintech, SA00001-1, dilution: 1:2000).

## Alanine-scan screening

A panel of synthetic peptides containing alanine substitution at all positions in the index HPV18E7$_{88-98}$ peptide was designed. This comprised 11 peptides were synthesized (Genscript, >90% pure). The response of 10F04mc transduced T cells toward K562-DR0901 cells pulsed with each peptide was determined by IFNγ Elisa. $1 \times 10^5$ T cells were incubated with $1 \times 10^5$ target cells in 96-well U-bottom plates for 16–24 h. The alanine mutated position was considered as a key recognition site when the IFNγ secretion induced by this alanine mutated peptide was less than 30% of HPV18E7$_{88-98}$.

## ELISA

For IFNγ or IL-2 secretion assays, $1 \times 10^5$ T cells were incubated with $1 \times 10^5$ target cells in 96-well U-bottom plates for 16–24 h. IFNγ or IL-2 secretion in the supernatants were measured by commercial ELISA kits (Mabtech, cat. 3420-1H-20, cat. 3445-1H-20) according to the manufacturer's instructions.

## Antibody blocking

For HLA blocking assays anti-Human HLA-A, B, C (W6/32, Biolegend, 311423), anti-Human HLA-DR, DP, DQ (Tu39, BD Biosciences, cat. 555556), anti-HLA-DR (G46-6, BD Biosciences, cat. 555809), anti-HLA-DP (B7/21, Abcam, cat. ab20897) and anti-HLA-DQ (SPL-L3, Abcam, cat. Ab23632) antibodies were added to the target cells at 50 μg/ml for 1 h prior to addition of the T cells. Anti-CD4 blocking experiment was performed as previously described with some modifications[51]. Neutralizing CD4 Monoclonal Antibody (RPA-T4, eBioscience, Cat. 16-0049-85) were added to TCR-T cells medium at 5 μg/ml 24 h prior to co-culture with target cells.

## Animal experiments

NOG (NOD.Cg-Prkdc$^{scid}$IL2rg$^{tm1Sug}$/JicCrl) mice (6–8 weeks old) were purchased from Beijing Vital River Laboratory Animal Technology Co., Ltd. and maintained in specific pathogen-free animal facilities at the housed in Guangzhou Curegenix Inc. In all experiments, mice were treated from 6 to 8 weeks old; and both males and females were used. All the experimental mice were housed under controlled temperature (21–23 °C) and 12:12 light: dark cycle conditions with free access to standard diet and water. Cervical tumor models were established by subcutaneous injection with $4 \times 10^6$/mice HeLa-DR0901 cells on day 1. A single intravenous injection of T cells (either TCR-T cells or Mock-T cells) was performed on day 7 following tumor cells injection. The dose of each experiment was described in the figure legends. The tumor volume was measured by caliper measurement of the

perpendicular diameters of each tumor using the formula: volume = (length × width$^2$)/2 every 3–5 days. Tumors were collected and weighed when sacrificed at the endpoint. Pharmacokinetic study of injected 10F04mc TCR-T cells were performed by JOINN Biologics Co. and carried out following the procedures of Good Laboratory Practice (GLP). Briefly, HeLa-DR0901 tumor-bearing mice were treated with 10F04mc TCR-T cells through i.v. injection. Mice (5 males and 5 females each time point) were euthanized at day 2, 8, 14, 21, 42, 63 for tissue collection (blood, tumor, spleen, lymph nodes). Genomic DNA (gDNA) were purified for absolute quantification PCR (qPCR) analysis.

## Statistical analysis

All statistical analyses were performed using GraphPad Prism version 8.0.1 (GraphPad Software, La Jolla, CA). Data are presented as means ± SD or means + SD, unless otherwise stated. The sample size for each experiment, $n$, is included in the results section and the associated figure legend. Statistical significance of differences between two groups was assessed by two-tailed Student's $t$ test. For all test, a $p$ value of <0.05 was considered significant.

## Reporting summary

Further information on research design is available in the Nature Portfolio Reporting Summary linked to this article.

## Data availability

All data featured in this paper are present in the paper or the Supplementary Materials. Source data are provided with this paper.

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

## Acknowledgements

The authors thank all patients and healthy donors for their consent of using their biopsy specimens and PBMC in this study. The study was sponsored by HRYZ Biotech Co. and supported by the Special funds of independent innovation industry development of Shenzhen Nanshan District (KC2015JSJS0015A).

## Author contributions

Y.H. supervised the entire study. Y.H., J.L., and J.C. designed the experiments. J.L., M.H., and Y.Z. collected and analyzed the human samples. X.C., S.O., M.M., and J.L. performed most of the experiments and analyzed data. Q.Y. performed the animal evaluation experiments. M.H., J.C., X.Q., and X.Z. provided important assistance or technique support. Y.H. and J.C. wrote and edited the manuscript.

## Competing interests

X.C., S.O., Q.Y., M.M., X.Z., J.C., and Y.H. are employees of HRYZ Biotech Co. All other co-authors declare no competing interests.
