## [Peer Review File · Nature Communications]

HLA-class II restricted TCR targeting human papillomavirus type 18 E7 induces solid tumor remission in miceREVIEWER COMMENTS

Reviewer #1 (Remarks to the Author):

Manuscript Nr: NCOMMS-23-32950-T

Long et al., "An HLA-class II restricted HPV18 E7 specific TCR cloned from a long-term surviving cervical cancer patient induces tumor remission in murine model"

The authors isolated a T cell receptor (TCR) from a patient with human papilloma virus 18 (HPV18) associated cervical carcinoma who stabilized disease for prolonged time periods after multiple antigen stimulating cellular therapy (MASCT). The TCR recognizes a peptide from the E7 HPV18 oncoprotein (aa88-98) presented on the MHC class II molecule HLA-DRA/DRB1*09:01. TCR transduced T cells, in particular those for which the TCR has been converted to a hybrid human-mouse molecule to avoid mispairing with the endogenous human TCR, recognize endogenously HPV18 E7 protein presented on HeLa cells that express HLA-DRA/DRB1*09:01 after transfection. When these cervical carcinoma cells were subcutaneously implanted into immune compromised mice adoptive transfer of TCR transgenic T cells reduced tumor growth with TCR transgenic T cell homing to the tumor microenvironment. Tumor control was better with TCR transgenic CD4+ than CD8+ T cells. An alanine scan of the cognate T cell epitope was performed to identify the important amino acids. Peptides predicted as cognate epitopes based on this motif were not efficiently recognized. Therefore, the authors argued that TCR transfer would not cause immunopathology. The TCR conferred both cytokine production and cytotoxicity to CD4+ and CD8+ T cells but in repeat stimulations TCR transgenic CD4+ T cells also performed better. MHC class II restricted recognition could also be improved by IFN-gamma. The authors were further able to demonstrate long-term persistence of this TCR in the original cancer patient. From these data the authors conclude that their isolated TCR might at least in part be responsible for the long disease stabilization in the patient from which they isolated it and should be explored for treatment of other HLA-DRA/DRB1*09:01 patients with HPV18 positive tumors.

These are interesting data, but it should be explored if the isolated TCR also recognizes E7 from other HPV types, what phenotype the respective TCR transgenic CD4+ and CD8+ T cells acquire in the tumor microenvironment, and at least some discussion why this TCR against HPV18 E7 might be more protective against associated malignancies than other HPV specific TCRs.

Major comments:

1. The authors find an interesting superiority of TCR transgenic CD4+ T cells and mixtures of CD4+ and CD8+ T cells in tumor rejection. They suggest that TCR transgenic CD8+ T cells exhaust faster

than CD4+ T cells while the latter maintain cytotoxicity. The authors provide little phenotypic characterization of the TCR transgenic T cells, especially in the tumor microenvironment in vivo, to support this hypothesis. Are indeed inhibitory receptors prominently upregulated on TCR transgenic CD8+ T cells in the tumor microenvironment? Are co-stimulatory receptors lost (CD27 and CD28) and senescence markers up-regulated? Are CD8+ T cells without CD4+ T cell help primarily TCF1 negative?

2. Two limitations might reduce the usefulness to apply the identified TCR clinically. One is that only HLA-DRA/DRB1*09:01 positive patients might benefit and the other that only HPV18 positive tumors might be targeted. While the authors addressed cross-reactivity with other MHC class II molecules, it would be interesting if E7 peptides from other HPV types might be recognized. They also might want to include subtypes that are not associated with cervical carcinoma to investigate if any of these might prime the investigated T cell specificity and thereby provide protection against the oncogenic subtypes.

3. Finally, the authors argue for superior protection by the investigated TCR but had initially isolated 5 TCRs against HPV18 E7 aa76-105. More information on these TCRs, their peptide specificity and inferior recognition of HLA-DRA/DRB1*09:01 transfected HeLa cells and possibly even primary HLA-DRA/DRB1*09:01 positive cervical carcinoma suspensions should be provided.

Minor comments:

1. The authors should discuss the frequency of HLA-DRA/DRB1*09:01 in the human population to estimate the number of cervical cancer patients that could benefit from the proposed adoptive TCR transgenic T cell therapy.

2. Even so the authors have previously published on their MASCT treatment, using monocyte-derived cytokine and polyI:C matured dendritic cells pulsed with tumor antigen derived peptides and subcutaneous injection, the manuscript would benefit from a brief description of the MASCT treatment.

3. The amount of transferred TCR transgenic T cells in the in vivo experiments of Figure 3E should be stated more clearly.

Reviewer #2 (Remarks to the Author):

The manuscript describes the isolation of mutant HPV18 E7-specific TCR (10F04) restricted to HLA class II from the blood of a long term surviving cervical cancer patient with a remarkable response to therapy. The TCR in question recognizes the epitope on HLA-DRB1*09:01 which has a frequency of

around 14-15% in the populations. The TCR was functional in both CD8 and CD4 T cells and demonstrated anti-tumour activity in vitro and in vivo. No cross-reactivity was detected when testing the core peptide sequence loaded onto LCLs, applying an alanine scan and testing HLA-DRB1*09:01 positive, HPV18 neg tumour cells. The TCR is now being in a phase I clinical trial for the treatment of HPV18-positive cancers (opened March 2023).

The study is very interesting and generally well-written. The TCR efficiency is well demonstrated, although not in several models, however, there are points that need clarification.

Major:

1. Figure 1c and Extended Figure 1 showing IFN-g ELISPOT responses against peptide pools in peripheral blood: How was the ELISPOT performed? What were the controls and were these ex vivo (no prestimulation responses)? It says one independent experiment with 4 time points assessed (baseline, 12, 15 and 24 months), so were other time points assessed during treatment?

It is explained in the figure legend of Figure 1c that the spot number displayed is the spot number for each peptide pool with the irrelevant peptide subtracted. Yet, values for ENV, indicated as the irrelevant peptide, are displayed as well. Please explain.

How many peptides were in each pool? More information is required for the methods for peptide pools and for ELISPOT kit. No information is given about the antibodies, manufacturer. Why were PBMCs stimulated twice in 48-hrs, was the stimulation repeated with the same peptide pools? How many cells were seeded per well and how many replicates were used for the ELISPOT plates?

2. The authors explain that 56 separate Va and Vb sequences were identified through scTCRseq, cloned and expressed, and 5 TCRs were found to be functional, recognizing HPV18E7 76-105 peptide. The data is not shown. Did the other 5 TCRs have the same HLA restriction? On which basis was TCR 10F04 selected (cytokine production, cytotoxicity)?

In Figure 2, the exact epitope is determined to be HPV18E7 84-98., but 88-102 is also recognized. Please show values of IFN-g measured in the ELISA assay. The HLA restriction is nicely determined, and then various cell lines are tested for recognition in Figure 2d. Did any of the non-LCLs endogenously express HLA-DRB1*09:01? HLA-modified HeLa cells with endogenous levels of HPV18 were also recognized. Still, cytokine production was very modest when tested in flow cytometry against K562 cells modified to express the appropriate HLA allele and loaded with peptide. TCR expression was not very high and thought to be due to inefficient pairing of the TCR chains in primary T cells (Extended Figure 2).

3. Murinization and codon optimization of the TCR highly increased the expression and function (Figure 3). The background of Granzyme B staining is very high with irrelevant peptide in Figure 3c, please explain. The plots show CD3+ T cells, was there any difference in functionality between TCR transduced CD4 and CD8 T cells? The killing efficiency shown in Figure 3d is high with low E:T ratios. Were total T cells used, both CD4 and CD8 T cells?

How was the killing efficiency calculated (which models was used)? How many replicates were tested per condition?

T cells were tested in an in vivo model using subcutaneous injection of HLA-modified HeLa cells. Did these HeLa cells only express endogenous levels of HPV18? What were the doses of T cells used? Please show statistics of differences in tumour load between groups in Figure 3e (not regression, please correct) and SD should be shown, not SEM.

Which dose of T cells was injected in Figure 3f?

In Figure 3g, it says patients, but only one test/primary tumour sample seems to be shown? Were any of the patients HPV+?

4. In Figure 4, cross-reactivity of the TCR was assessed using IFN-g ELISAs testing T cells against LCLs loaded with the core peptide sequence through an alanine scan screening. Additionally a BLAST search identified peptides derived from human proteins that could possibly be recognized, and these were synthesized and tested. No HPV18 negative, HLA-DRB1*09:01 positive cancer cell lines were recognized. Did all these cell lines express reasonable levels of the HLA-DR molecules or were they only genotyped to be HLA-DRB1*09:01 positive?

5. In Table 1, tracking of the 10F04 TCR was performed in the cervical patient's blood at different time points. Please explain which sequencing depths were used? Very limited methods section for bulk TCR-seq, please expand.

6. In figure 5, the CD4 and CD8 T cells are tested for functionality separately. Please change the term pluripotent for multifunctional in the title of the figure. The CD4 TCR-modified T cells produce less IFN-g than the CD8 T cells, but more of the other cytokines and Granzyme B. A higher percentage of CD4 effector memory cells were found in the CD4 T cell population compared to the CD8 T cell population when looking at phenotype. Why would this be? The killing capacity of CD4 T cells after multiple rounds of re-challenge with target cells (HLA-modified HeLa cells) was shown to be higher in Figure 5c, and upon adoptive transfer of either TCR-modified CD4 T cells or CD8 T cells the same was shown. Indeed, CD8 T cells were largely ineffective compared to mock T cells. Why is this? Did the two T cell populations have similar TCR expression levels? If the TCR-modified CD8 T cells kill target cells in the first round, why would simply blocking CD4 be sufficient to inhibit killing in Figure 5e?

In Figure 5e, the CD4 T cell controls do not kill as efficiently as the mix of CD4/CD8 T cells without anti-CD4 whereas the CD8 T cells alone do. Please explain.

The authors could also look at TCR-modified T cells in vivo and see if CD4 T cells are the ones proliferating when a mix of CD4 and CD8 T cells are injected.

It should be discussed in the manuscript why this difference in efficacy between CD4 and CD8 T cells occurs so quickly both in vitro and in vivo. Were any phenotypic markers looked at after rechallenge? CD4 T cells can definitely be efficient cytotoxic cells, but this difference is very striking between the two. What happens to the CD8 T cells? This should be investigated further by flow cytometry phenotyping and functional assays.

7. The methods section is lacking a lot of details (some mentioned above). For flow cytometry, please provide more details on kits and antibodies. Clones should be added (consider a table), and it says anti-mouse TCR antibodies were used, please correct. Please provide more details on antibodies for ELISAs. For in vivo assays, T cell dose is missing. What were humane endpoints in the in vivo studies, i.e. when were mice sacrificed? Indications for statistical testing are missing in several figures.

Minor

Cite more of the original papers when referring to specific clinical studies of TCR treatment from review PMID: 36791198 of five clinical trials for NY-ESO-1 TCR, and consider including a recent study: PMID: 37586317.

Please change the word pluripotency with multifunctional for T cells secreting multiple cytokines. Pluripotency refers to the capacity of individual cells to initiate all lineages.

In Figure 1a, correct survive, should be survival

Check sentences and typos, e.g. line 41: "...is the lacking pre-existing T cells" should be "is the lack of...", line 74: long-term surviving patients rather than long-term survived, line 82: well-immune-response should be corrected,

POINT-BY-POINT RESPONSE TO REVIEWERS' COMMENTS

REVIEWER COMMENTS

Reviewer #1 (Remarks to the Author):

These are interesting data, but it should be explored if the isolated TCR also recognizes E7 from other HPV types, what phenotype the respective TCR transgenic CD4⁺ and CD8⁺ T cells acquire in the tumor microenvironment, and at least some discussion why this TCR against HPV18 E7 might be more protective against associated malignancies than other HPV specific TCRs.

Response: The authors are very grateful for recognizing the value of our work and helping us improve the manuscript. In light of your valuable suggestions, we conducted additional experiments to address your questions and concerns. Detailed point-to-point responses are shown below.

Major comments:

1. The authors find an interesting superiority of TCR transgenic CD4⁺ T cells and mixtures of CD4⁺ and CD8⁺ T cells in tumor rejection. They suggest that TCR transgenic CD8⁺ T cells exhaust faster than CD4⁺ T cells while the latter maintain cytotoxicity. The authors provide little phenotypic characterization of the TCR transgenic T cells, especially in the tumor microenvironment in vivo, to support this hypothesis. Are indeed inhibitory receptors prominently upregulated on TCR transgenic CD8⁺ T cells in the tumor microenvironment? Are co-stimulatory receptors lost (CD27 and CD28) and senescence markers up-regulated? Are CD8⁺ T cells without CD4⁺ T cell help primarily TCF1 negative?

Response: We would like to express our gratitude for the positive recognition of our discovery by the reviewer. In order to further elucidate the functional and phenotypic variances between CD4 TCR-T and CD8 TCR-T cells, we conducted multiple rounds of co-culture with tumor target cells and monitored various indicators. We observed that CD8 TCR-T cells exhibited rapid killing and expansion in the initial co-culture, followed by a decline in cell numbers and IFN γ secretion in subsequent rounds (**Figure 5c, d, Supplementary Figure 9b, c**). In contrast, CD4 TCR-T cells consistently demonstrated sustained killing efficacy, maintained high cell

numbers, and consistently secreted IFN γ throughout the multiple rounds of co-culture (**Figure 5c, d, Supplementary Figure 9b, c**). Additionally, our analysis of cell phenotype revealed distinctive expression patterns of CD27 and CD39 in the two T cell subtypes, indicating potential differences in functionality. Other activation and exhaustion markers, such as CD28, CD69, PD1, TIM3, and TIGIT, did not show significant differences between CD4 and CD8 TCR-T cells (**Supplementary Figure 9d, Figure R1**). Studies have suggested that the high expression of CD39 is related to weaker T cell tumor-killing function^{1,2}. Therefore, the initially low expression of CD39 in CD4 TCR-T cells may be related to their sustained functionality.

The present immunodeficient xenograft mouse models lack human immune cells and may not be an ideal model for the phenotypic characterization of TCR-T cells. As this TCR-T therapy has advanced to the clinical stage and the initial infusions have been completed, we aim to proceed with translational medical research in forthcoming clinical trials to further explore the phenotypic changes of different TCR-T cell subtypes in real human tumor microenvironments and investigate their association with therapeutic efficacy.

Supplementary Figure 9. Characterization of CD4 and CD8 10F04mc TCR-T cells after

repeat challenges with tumor cells.

b. Killing ration summary of 4 independent repeat challenge experiment in **Figure 5c**.

c. 10F04mc transduced CD4⁺TCR-T cells produced more IFN γ than CD8⁺TCR-T cells in serial killing. IFN γ secretion was detected by ELISA.

d. The expression of the cell surface markers CD27, CD39 and PD1 on 10F04mc transduced CD4⁺ TCR-T cells and CD8⁺ TCR-T cells in serial killing.

Data represent three independent experiments performed in technical replicates. Error bars indicate means \pm SD (**a-c**).

Figure R1. The expression of the activation markers CD28, CD69 and exhaustion markers CD39, TIM3, TIGIT on 10F04mc transduced CD4⁺ TCR-T cells and CD8⁺ TCR-T cells in serial killing.

2. Two limitations might reduce the usefulness to apply the identified TCR clinically. One is that only HLA-DRA/DRB1*09:01 positive patients might benefit and the other that only HPV18 positive tumors might be targeted. While the authors addressed cross-reactivity with other MHC class II molecules, it would be interesting if E7 peptides from other HPV types might be recognized. They also might want to include subtypes that are not associated with cervical carcinoma to investigate if any of these might prime the investigated T cell specificity and thereby provide protection against the oncogenic subtypes.

Response: We are grateful for the invaluable recommendation provided by the reviewer. In response, we conducted a comprehensive comparison of HPV E7 protein sequences across all subtypes in the database and synthesized 11 peptides with the highest similarity to the HPV18 E7₈₈₋₉₈ epitope (**Supplementary Figure 7a**). Through peptide recognition experiments, we found that the 10F04mc TCR-T cells also recognize the HPV45 E7₈₉₋₉₉ epitope

(QLFLSTLSFVC), whereas the recognition of the HPV85 E7₉₁₋₁₀₁ epitope (QLFLGTLFSLC) is notably weak (Supplementary Figure 7b). Despite sharing the same core epitope with HPV18 E7₈₈₋₉₈, HPV45 E7₈₉₋₉₉, and HPV85 E7₉₁₋₁₀₁ exhibit 1 or 2 different amino acids. The avidity of 10F04mc TCR-T against HPV45 E7₈₉₋₉₉ is slightly weaker compared to the HPV18 E7₈₈₋₉₈ epitope (Supplementary Figure 7c).

Despite the HPV45 infection is also associated with cervical cancer, the infection rate is much lower than the HPV18 virus. HPV45 virus was found in approximately 0.5% of the general population and about 5% of cervical cancer patients³. The recognition of the HPV45E7 antigenic epitope by 10F04mc TCR-T suggests the potential therapeutic benefits for patients with HPV45⁺ tumors from 10F04mc TCR-T therapy. Please accept our gratitude for offering constructive insights, which have further enriched our work.

Supplementary Figure 7. Cross-recognition analysis of 10F04mc TCR-T cells against

HPV-derived epitopes.

a. HPV family sequence homology between the E7 proteins in the Core epitope of 10F04mc TCR.

b. 10F04mc TCR cross-reactive to HPV47E7. 10F04mc transduced T cells were stimulated by LCLs pulsed with HPV18E7₈₈₋₉₈, HPV45₈₉₋₉₉, HPV70₉₁₋₁₀₁, HPV57₇₈₋₈₈, HPV85₉₁₋₁₀₁, HPV59₉₀₋₁₀₀, HPV221₇₂₋₈₂, HPV68₉₂₋₁₀₂, and HPV39₉₁₋₁₀₁. The concentration of IFN γ in the supernatant after overnight co-culture was detected by ELISA.

c. The binding avidity of 10F04mc TCR to HPV18E7₈₈₋₉₈ and HPV45₈₉₋₉₉. IFN γ secretion was detected by ELISA.

Data represent three independent experiments performed in technical replicates. Error bars indicate means \pm SD (**b-c**).

3. Finally, the authors argue for superior protection by the investigated TCR but had initially isolated 5 TCRs against HPV18 E7 aa76-105. More information on these TCRs, their peptide specificity and inferior recognition of HLA-DRA/DRB1*09:01 transfected HeLa cells and possibly even primary HLA-DRA/DRB1*09:01 positive cervical carcinoma suspensions should be provided.

Response: Thanks for the comment. Following the recommendations, we would like to share detailed research insights on the five TCRs. All the five TCRs were subsequently transduced into human T cells. The result revealed variations in the expression efficiency of most of the TCR molecules, along with differential multifactorial expression of IFN γ and TNF α following stimulation with the HPV18 E7₇₆₋₁₀₅ antigen peptide-loaded autologous LCL (**Supplementary Figure 2a, b**). While all five TCR molecules exhibited some level of recognition of the HPV18 E7₇₆₋₁₀₅ antigen, the most optimal expression and recognition capabilities were observed in the 10F04 and P09B08 TCR-transduced T cells. Furthermore, our study unveiled that the core epitope sequence recognized by the P09B08 TCR is identical to that of 10F04, both targeting the HPV18 E7₈₈₋₉₈ epitope (QLFLSTLSFVC) (**Figure R2a**), with the same HLA restriction as HLA-DRB1*09:01 (**Figure R2b, c**). However, the affinity of the P09B08 TCR molecule was found to be weaker than that of the 10F04 TCR (**Figure R2d**). Consequently, in consideration of the overall performance, we have chosen to proceed with the 10F04 TCR for

subsequent research and development among the five TCR molecules.

Supplementary Figure 2. Characterization of HPV18 E7 reactive TCRs.

a. Flow cytometry analysis of TCR expression in T cells transduced HPV18 E7 reactive TCRs 33A02,09B12 and P09B08, only the 33D05 TCR flow cytometry antibody is not available.

b. The HPV18 E7 reactive TCRs transduced T cells produced TNF α and IFN γ after *in vitro* stimulation with autologous LCL pulsed with HPV18E7₇₆₋₁₀₅.

Figure R2. Characterization of HPV18 E7 reactive TCRs P09B08

a. Epitope identification of P09B08. The P09B08 transduced T cells secreted a great amount of

IFN γ after in vitro stimulation of autologous LCL pulsed with the HPV18E7₇₆₋₁₀₅. Series truncated peptides derived from HPV18E7₇₆₋₁₀₅ were pulsed on autologous LCLs to stimulate P09B08 transduced T cells. HPV18E7₈₄₋₁₀₂ was identified as the epitope of P09B08. The concentration of IFN γ in the supernatant after overnight co-culture was detected by ELISA.

b. HLA restriction identification of P09B08. The P09B08 transduced T cells were co-cultured with autologous LCLs pulsed with the HPV18E7₇₆₋₁₀₅ peptide in the presence of blocking antibodies or isotype control. The IFN γ secretion was detected by ELISA and normalized to isotype control. Only the HLA-DR antibody showed a complete blocking.

c. P09B08 transduced T cells were co-cultured with HEK-293T cells expressing each HLA-II molecule of the autologous LCL, which were pulsed with HPV18E7₇₆₋₁₀₅ peptide. Autologous LCLs pulsed with HPV18E7₇₆₋₁₀₅ peptide were used as a positive control. The HLA restriction of P09B08 was determined as HLA-DRB1*09:01. IFN γ secretion was detected by ELISA.

d. HPV18E7₇₆₋₁₀₅ peptide titration were used to assess the binding avidity of 10F04 and P09B08. 10F04 transduced T cells showed a stronger binding avidity. IFN γ secretion was detected by ELISA.

Data represent three independent experiments performed in technical replicates. Error bars indicate means \pm SD (**a-d**).

Minor comments:

1. The authors should discuss the frequency of HLA-DRA/DRB1*09:01 in the human population to estimate the number of cervical cancer patients that could benefit from the proposed adoptive TCR transgenic T cell therapy.

Response: We thank the reviewer for the suggestion, and the HLA-DRA/DRB1*09:01 frequency information has been added to the revised manuscript. The HLA-DRA/DRB1*09:01 allele is predominantly found in the Asian population. According to the data from the Allele Frequency Net Database (AFND, <http://allelefrequencies.net>), the individuals that have the HLA-DRA/DRB1*09:01 allele approximately 20.43% in the overall Asian population (n=1772, allele frequency =10.80%), 28.67% in the Chinese population (n=99672, allele frequency =28.67%), 25.82% in the Japanese population (n=24582, allele frequency =13.87%), and 18.40% in the Korean population (n=77584, allele frequency =9.67%). These statistics suggest that a

significant number of cervical cancer patients in these regions may benefit from the proposed adoptive TCR transgenic T cell therapy.

2. Even so the authors have previously published on their MASCT treatment, using monocyte-derived cytokine and polyI:C matured dendritic cells pulsed with tumor antigen derived peptides and subcutaneous injection, the manuscript would benefit from a brief description of the MASCT treatment.

Response: We thank the reviewer for the suggestion, and the reviewer's point is well taken. A brief description about MASCT treatment has been added in the revised manuscript.

3. The amount of transferred TCR transgenic T cells in the in vivo experiments of Figure 3E should be stated more clearly.

Response: Thank you very much for your comments. We have added the dose information in the figure legend of Figure 3. TCR-T groups received a single intravenous injection at a different dose (Low: 1×10^7 , medium: 3×10^7 , High: 5×10^7 transduced T cells per mouse, respectively) 6 days after the tumor cells injection. Control group mice received 5×10^7 Mock-T cells at the same day.

Reviewer #2 (Remarks to the Author):

The manuscript describes the isolation of mutant HPV18 E7-specific TCR (10F04) restricted to HLA class II from the blood of a long term surviving cervical cancer patient with a remarkable response to therapy. The TCR in question recognizes the epitope on HLA-DRB1*09:01 which has a frequency of around 14-15% in the populations. The TCR was functional in both CD8 and CD4 T cells and demonstrated anti-tumour activity in vitro and in vivo. No cross-reactivity was detected when testing the core peptide sequence loaded onto LCLs, applying an alanine scan and testing HLA-DRB1*09:01 positive, HPV18 neg tumour cells. The TCR is now being in a phase I clinical trial for the treatment of HPV18-positive cancers (opened March 2023).

The study is very interesting and generally well-written. The TCR efficiency is well demonstrated, although not in several models, however, there are points that need clarification.

Response: We thank this reviewer for his excellent summary and the positive evaluation of our work. According to the constructive comments and suggestions, we have carried out a substantial amount of new experiments to address your concerns.

Continuing from the HeLa cell model, we have expanded our scope to encompass a variety of cell models for both *in vitro* and *in vivo* experiments. These models include endogenous HPV18-positive human cervical cancer cells C-4I and human squamous cell carcinoma SW756, as well as HPV18-negative human cervical cancer cell lines MS751 and human osteosarcoma cell line U2OS. The *in vitro* recognition and cytotoxicity assays have demonstrated the specificity of 10F04mc TCR-T target recognition and its effective killing function against HLA-DRB1*09:01⁺& HPV18E7⁺ tumor cells (**Supplementary Figure 4a-i**). Furthermore, in another *in vivo* model, we validated the superior anti-tumor efficacy of 10F04mc TCR-T against HPV18 E7 protein endogenously expressed HeLa-DR0901 cells (**Supplementary Figure 3d, e**) Additionally, we have further analyzed the differential characteristics and potential mechanisms of CD4⁺ TCR-T and CD8⁺ TCR-T cell functions. The aforementioned results have been incorporated into the revised manuscript. Detailed point-to-point responses are shown below.

Supplementary Figure 4. Antitumor activity of 10F04mc TCR-T cells in different cell models.

a. *In vitro* cytotoxicity assay of endogenous HPV18-positive human cervical cancer cells c-4I and SW756 (c), HLA-DRB1*09:01transduced C-4I and SW756 cells by 10F04 TCR-T cells. HPV18 E7 protein expression was been validated by western blotting (b, d).

e. *In vitro* cytotoxicity assay of HPV18-negative human cervical cancer cells MS751 and human

osteosarcoma cell line U2OS (c), HLA-DRB1*09:01 and HPV18 E7 protein double transduced MS751 and U2OS cells by 10F04 TCR-T cells. HPV18 E7 protein expression was been validated by western blotting (f, h).

i. Flow cemetery analysis of HLA-DRB1*09:01 expression of those cell lines.

Major:

1a. Figure 1c and Extended Figure 1 showing IFN-g ELISPOT responses against peptide pools in peripheral blood: How was the ELISPOT performed? What were the controls and were these ex vivo (no prestimulation responses)? It says one independent experiment with 4 time points assessed (baseline, 12, 15 and 24 months), so were other time points assessed during treatment?

Response: The IFN γ -ELISPOT assay was performed by commercial kits including two rounds of ex vivo stimulation with TAA peptides or HPV antigen peptides. For the first round, patients' PBMCs were stimulated with peptides for 48 hours in 96-well plates. Thereafter cells were transferred onto 96-well ELISPOT assay plates and stimulated again with the same peptides following by the IFN γ detection. As a negative control, patients' PBMCs were stimulated with irrelevant peptides (HIV envelope peptide, ENV) or no peptides (W/O). As a positive control, patients' PBMCs were stimulated with peptide pools (including all detected peptides).

Due to the constraints in obtaining a limited amount of patient PBMCs for ELISPOT assays, each immune monitoring point only included a single testing, with three replicate wells designated for each antigen in every trial. Meanwhile, with the same reason, immune monitoring for the patient was not performed 2 years after initial treatment. Instead, we adopted TCR-seq to detect the presence of 10F04 TCR $\alpha\beta$ chains in her peripheral blood samples collected at 2 years post-initial treatment (2016), 3 years (2017), 4 years (2018), and 7 years (2021) (Table 1).

1b. It is explained in the figure legend of Figure 1c that the spot number displayed is the spot number for each peptide pool with the irrelevant peptide subtracted. Yet, values for ENV, indicated as the irrelevant peptide, are displayed as well. Please explain.

Response: Thanks for your helpful comment. The data presented in Figure 1c and

Supplementary Figure 1 were the original spot number 2×10^5 PBMCs without subtracting the spot number obtained from the irrelevant peptide control. We apologize for the confusion and the related description in the figure legend have been corrected.

1c. How many peptides were in each pool? More information is required for the methods for peptide pools and for ELISPOT kit. No information is given about the antibodies, manufacturer. Why were PBMCs stimulated twice in 48-hrs, was the stimulation repeated with the same peptide pools? How many cells were seeded per well and how many replicates were used for the ELISPOT plates?

Response: Thanks for your comment. We apologize for the lack of detailed information in the method section. Peptide pools contained all of the stimulated peptides, but the concentration of each peptide used is one-fifth of a single peptide (single peptide: 10ug/mL, each peptide in the peptide pools: 2ug/mL). The commercial IFN γ -ELISPOT kits were produced by U-CyTech biosciences (U-CyTech, cat. 230PR5) of the Netherlands (Europe). Unfortunately, information about the antibodies were not provided in the instructions.

Peptides used in the IFN γ -ELISPOT assay were long peptides (about 30 AAs). Long peptides must first be internalized, processed, and presented by APCs via MHC class I/II molecules before they can stimulate cytokine (or other effector molecule) release by T cells, according to the manufacturer's instructions. Therefore, PBMCs must be stimulated twice in 48 hours. 3×10^5 PBMCs were seeded per well and 3 replicates were used for the ELISPOT plates. The method section has also been updated.

2a. The authors explain that 56 separate Va and Vb sequences were identified through scTCRseq, cloned and expressed, and 5 TCRs were found to be functional, recognizing HPV18E7 76-105 peptide. The data is not shown. Did the other 5 TCRs have the same HLA restriction? On which basis was TCR 10F04 selected (cytokine production, cytotoxicity)?

Response: Thanks for your comment. Please refer to the response of the Major comments 3 from reviewer #1.

2b. In Figure 2, the exact epitope is determined to be HPV18E7 84-98., but 88-102 is also recognized. Please show values of IFN-g measured in the ELISA assay.

Response: We thank the reviewer for the suggestion. The figure has been revised accordingly.

2c. The HLA restriction is nicely determined, and then various cell lines are tested for recognition in Figure 2d. Did any of the non-LCLs endogenously express HLA-DRB1*09:01? HLA-modified HeLa cells with endogenous levels of HPV18 were also recognized.

Response: Thanks for the comment. HLA-II molecules are mainly expressed on the surface of immune cells, such as antigen-presenting cells and B cells. Lymphoblastoid cell lines (LCLs) derived from B cells are known to express HLA-II molecules, as confirmed by HLA-DR flow cytometry and sequencing. Contrarily, tumor cell lines like Caski and HeLa are generally lack HLA-II molecule expression. However, through artificial overexpression of HLA-DRB1*09:01, successful HLA-DR expression in these cell lines was verified via HLA-DR flow cytometry staining (**Figure R3**).

Figure R3. HLA-DR expression validation by flow cytometry.

2d. Still, cytokine production was very modest when tested in flow cytometry against K562 cells modified to express the appropriate HLA allele and loaded with peptide. TCR expression was not very high and thought to be due to inefficient pairing of the TCR chains in primary T cells (Extended Figure 2).

Response: The inherent potential for the mispairing of the transduced wild-type TCR and the endogenously expressed TCR can lead to a lower TCR positivity rate and subsequent attenuation of TCR functionality. As a strategic response, we replaced the constant regions of

the 10F04 TCR with the murine constant regions in our subsequent experiments to mitigate mispairing. Encouragingly, as demonstrated in **Figures 3a** and **3b**, the murinization of the TCR yielded a substantial increase in TCR expression and target recognition (**Supplementary Figure 2a, b**). These results underscore the critical role of TCR engineering in improving both the quantity and quality of TCR-T cells, thereby enhancing their functional capabilities.

3a. Murinization and codon optimization of the TCR highly increased the expression and function (Figure 3). The background of Granzyme B staining is very high with irrelevant peptide in Figure 3c, please explain.

Response: Thanks for the comment. In the process of preparing TCR-T, T cells were activated using anti-CD3 and anti-CD28 antibodies. The activation leads to a background of non-specific Granzyme B expression in T cells. Our further analysis found that the Granzyme B positive staining background mainly comes from CD8⁺ T cells (**Supplementary Figure 8**). Several previous studies have reported that CD8⁺ T cells are the primary subset responsible for producing Granzyme B, which aligns with the results we detected^{4,5}.

3b. The plots show CD3⁺ T cells, was there any difference in functionality between TCR transduced CD4 and CD8 T cells?

Response: Thank you for your comment. We have conducted further analysis of the multifunctional features of the CD4⁺ TCR-T and CD8⁺ TCR-T cells. Our results demonstrate that following a short period of target antigen stimulation (4 hours), both CD4⁺ TCR-T and CD8⁺ TCR-T cells express the three cytokines IFN-gamma, IL-2, and TNF-alpha. Notably, CD4⁺ TCR-T cells show a robust expression of TNF-alpha and IL-2 shortly after stimulation, while the number of IFN-gamma-positive cells is relatively lower compared to CD8⁺ TCR-T cells. Conversely, CD8⁺ TCR-T cells exhibit a significant expression of IFN-gamma shortly after stimulation. Furthermore, Granzyme B is upregulated in both CD4⁺ TCR-T and CD8⁺ TCR-T cells, with CD8⁺ TCR-T cells displaying higher baseline expression prior to antigen-specific stimulation (**Supplementary Figure 8**). These findings strongly indicate that CD4⁺ and CD8⁺ cells transduced with 10F04mc TCR exhibit multifunctionality, although the specific expression levels of functional factors and response speed may vary.

Supplementary Figure 8. Multiple cytokines expression of 10F04mc TCR transduced T cells after stimulated with HPV18E7₇₆₋₁₀₅ peptide.

The 10F04mc transduced T cells produced IFN γ , IL-2, TNF α and Granzyme B after *in vitro* stimulation. 10F04mc transduced T cells were co-cultured with K562-DRA/DRB1*09:01 pulsed with HPV18E7₈₄₋₉₈ or irrelevant peptide. Intracellular IFN γ , IL-2, TNF α , and Granzyme B production were analyzed by gating in the CD4⁺ T cells and CD8⁺ T cells.

3c. The killing efficiency shown in Figure 3d is high with low E:T ratios. Were total T cells used, both CD4 and CD8 T cells?

Response: Thank you for your question. The experimental setup entailed keeping the number of target cells constant while introducing different quantities of T cells based on various effector-to-target ratios. The T cell population used was total T cells, without specific isolation

of CD4, CD8, or TCR+ cells. Each well contained 8000 target cells.

3d. How was the killing efficiency calculated (which models was used)? How many replicates were tested per condition?

Response: Thank you for the question. The killing efficiency assay for each condition involved triplicate testing. We utilized the Agilent xCELLigence RTCA system to evaluate the cell viability of target cells. The functional unit of a cellular impedance assay is a set of gold microelectrodes fused to the bottom surface of a microtiter plate well. When submersed in an electrically conductive solution (such as standard tissue culture medium), the application of an electric potential across these electrodes causes electrons to exit the negative terminal, pass through bulk solution, and then deposit onto the positive terminal to complete the circuit. Because this phenomenon is dependent upon the electrodes interacting with bulk solution, the presence of adherent cells at the electrode-solution interface impedes electron flow. The magnitude of this impedance is dependent on the number of cells, the size and shape of the cells, and the cell-substrate attachment quality. The killing efficiency was calculated using the formula: $\% \text{ killing} = (\text{Normalized Cell index}_{\text{Mock-T}} - \text{Normalized Cell index}_{\text{TCR-T}}) / \text{Normalized Cell index}_{\text{Mock-T}} \times 100$ at the endpoint time of measurement.

3e. T cells were tested in an in vivo model using subcutaneous injection of HLA-modified HeLa cells. Did these HeLa cells only express endogenous levels of HPV18? What were the doses of T cells used? Please show statistics of differences in tumour load between groups in Figure 3e (not regression, please correct) and SD should be shown, not SEM.

Response: In the previous version of the manuscript, the *in vivo* xenograft experiments utilized HPV18 E7 protein overexpressed HeLa cells (HeLa-DR0901-HPV18E7). In the revised version, we have included the results of *in vivo* and *in vitro* results using HPV18 E7 endogenous expressed tumor model HeLa cells (HeLa-DR0901) (**Supplementary Figure 3a, b**). The results indicate that 10F04mc TCR-T exhibited similar killing efficacy against both cell lines in the *in vitro* cytotoxicity assay (**Supplementary Figure 3c**). Consistently, *in vivo*, 10F04mc TCR-T showed significant inhibitory effect on the HPV18 E7 endogenous expressed HeLa cell line. However, at the endpoint, a higher proportion of mice in the high-dose treatment group

with HPV18E7-overexpressing cells achieved complete tumor clearance (2/6 vs 0/5) (Supplementary Figure 3d, e).

We have completed the methods section and figure legends of the revised manuscript to include all missing information, including details on the T cell dosage. Statistical analysis and figures have been updated as suggested.

Supplementary Figure 3. Antitumor efficiency of 10F04mc TCR-T cells against tumor cells with different expression levels of HPV18 E7 protein.

a. Western blotting analysis of HPV18E7 in HeLa, HeLa-DR0901 and HeLa-DR0901-HPV18E7 cells.

b. HLA-DR expression analysis of HeLa, HeLa-DR0901 and HeLa-DR0901-HPV18E7 cells.

c. *In vitro* killing assay of HeLa, HeLa-DR0901 and HeLa-DR0901-HPV18 cells by 10F04

transduced T cells. 10F04 transduced T cells can specifically kill HeLa-DR0901 and HeLa-DR0901-HPV18 cells.

d. *In vivo* antitumor activity of 10F04mc TCR-T cells in HPV18 E7 endogenous expressed tumor model. NOG mice were injected subcutaneously with 4×10^6 HeLa-DR0901 cells per mouse at Day 1. TCR-T groups received a single intravenous injection at a different dose (Low: 1×10^7 , High: 5×10^7 transduced T cells per mouse, respectively) 6 days after the tumor cells injection. Control group mice received 5×10^7 Mock-T cells at the same day. The tumor volume was measured by digital caliper every 2-5 days. Mice were euthanized at Day 48 for tumor isolation and weighing (**e**).

3f. Which dose of T cells was injected in Figure 3f?

Response: Thank you very much for your comments. We have added the dose information in the figure legend of **Figure 3**. TCR-T groups received a single intravenous injection at a different dose (Low: 1×10^7 , medium: 3×10^7 , High: 5×10^7 transduced T cells per mouse, respectively) 6 days after the tumor cells injection. Control group mice received 5×10^7 Mock-T cells on the same day.

3g. In Figure 3g, it says patients, but only one test/primary tumour sample seems to be shown? Were any of the patients HPV+?

Response: Thank you for pointing out the writing error. We apologize for the confusion. Figure 3g presents three independent experiment results of three donors. Regrettably, all three patients tested negative for HPV18. Consequently, we have employed exogenous loading of HPV18E7 antigen peptides as a strategy to simulate HPV18⁺ tumor tissue in this study.

4. In Figure 4, cross-reactivity of the TCR was assessed using IFN-g ELISAs testing T cells against LCLs loaded with the core peptide sequence through an alanine scan screening. Additionally a BLAST search identified peptides derived from human proteins that could possibly be recognized, and these were synthesized and tested. No HPV18 negative, HLA-DRB1*09:01 positive cancer cell lines were recognized. Did all these cell lines express

reasonable levels of the HLA-DR molecules or were they only genotyped to be HLA-DRB1*09:01 positive?

Response: Thank you for the comment. HLA type of all utilized LCLs and cell lines have been identified using the sequencing-based typing (SBT) method. Additionally, flow cytometry was employed to assess the cell surface expression of their HLA-DR molecules.

5. In Table 1, tracking of the 10F04 TCR was performed in the cervical patient's blood at different time points. Please explain which sequencing depths were used? Very limited methods section for bulk TCR-seq, please expand.

Response: Thank you very much for your comments. We have provided a more detailed description in the Method section of the manuscript. The bulk TCR NGS-library is performed by Multiplex PCR, with a size ranging from 470-500 bp. In **Table 1**, each read represents a different TCR sequence, rather than a fragment of the genome. Therefore, we do not calculate sequencing depths but instead calculate "Reads coverage per cell". At least one million high-quality reads were collected for one million loaded PBMCs. Theoretically, The TCR sequence from each T cell can be covered at least once.

6a. In figure 5, the CD4 and CD8 T cells are tested for functionality separately. Please change the term pluripotent for multifunctional in the title of the figure.

Response: Thank you very much for your careful review. Based on your comment, we have replaced the word.

6b. The CD4 TCR-modified T cells produce less IFN-g than the CD8 T cells, but more of the other cytokines and Granzyme B. A higher percentage of CD4 effector memory cells were found in the CD4 T cell population compared to the CD8 T cell population when looking at phenotype. Why would this be?

Response: Thanks for the question. The result of **Figure 5b** reveals that CD4⁺ TCR-T cells present a higher proportion of Tem phenotype, while CD8⁺ TCR-T cells are primarily Teff. Effector cells typically respond rapidly upon encountering target cells, while memory cell responses require more time. The flow cytometry experiment in **Figure 5a**, assessing

intracellular cytokine expression 4 hours post-stimulation with APC cells, demonstrates higher IFN γ expression in CD8⁺ TCR-T cells due to the shorter stimulation period. In contrast, the results from **Supplementary Figure 9a**, measuring IFN γ levels in cell culture supernatants 24 hours post-stimulation, indicate that CD4⁺ TCR-T cells can secrete more cytokines after prolonged stimulation. These findings are consistent with the ELISA results from the continuous killing experiment shown in **Supplementary Figure 9c**. Previous studies have reported similar cytokine expression patterns; the expression of IFN γ was favored in CD8 T cells, while TNF α and IL-2 showed a preference for CD4 T cells.

As the data shown in **Supplementary Figure 8**, Granzyme B expression is much higher in activated CD8⁺ TCR-T cells without stimulation. The result presented in **Figure 5a** show the changes in expression levels after background subtraction. Consequently, due to the lower background levels in CD4⁺ TCR-T cells, the upregulation of Granzyme B expression after activation is more pronounced compared to CD8⁺ TCR-T cells.

6c. The killing capacity of CD4 T cells after multiple rounds of re-challenge with target cells (HLA-modified HeLa cells) was shown to be higher in **Figure 5c**, and upon adoptive transfer of either TCR-modified CD4 T cells or CD8 T cells the same was shown. Indeed, CD8 T cells were largely ineffective compared to mock T cells. Why is this? Did the two T cell populations have similar TCR expression levels? If the TCR-modified CD8 T cells kill target cells in the first round, why would simply blocking CD4 be sufficient to inhibit killing in **Figure 5e**?

Response: In the cytotoxicity assay, both CD4 T and CD8 T cells exhibited initial killing capacities against target cells with no significant difference. However, as the rounds of cytotoxicity increased, the cytotoxic activity of CD8 T cells gradually declined to almost negligible levels by the fourth round, while CD4 T cells sustained their cytotoxic capabilities against tumor cells. This observation, combined with data from **Figure 5b**, suggests that CD4 T cells may have a larger reservoir of memory-type cells, capable of sustained tumor cytotoxicity through differentiation into effector cells. Constantly, we observed more CD4 TCR-T cells exist during the repeat tumor cell challenge, while CD8 TCR-T cells were dropped quickly after the 2nd round challenge (**Figure 5d**).

The transduction efficiency of CD4⁺ T cells is slightly higher compared to CD8 T cells in some donors. Nevertheless, the expression of 10F04mc TCR on the surface of CD4⁺ and CD8⁺ cells are generally comparable (**Figure 3a**).

Figure 3a. Flow cytometry analysis of 10F04 expression in CD4 and CD8 human T cells. 10F04 after murinization in the C region and codon optimization (10F04mc) was nicely expressed on both CD8⁺ and CD4⁺ T cells after lentivirus transduction as compared to non-transduced T cells. Right: Collection of 6 independent experiment result of 6 donors derived T cells.

In **Figure 5f**, the cytotoxicity assay utilized a low E : T ratio (2:1), with bulk CD3⁺ TCR-T cells and purified CD4⁺ TCR-T or CD8⁺ TCR-T cell subpopulations. The CD3⁺ TCR-T group included an equivalent total number of loading TCR-T cells as the other two groups. However, due to its composite cellular composition (with ~70% CD8⁺), the cytotoxicity of the CD3⁺ TCR-T group exhibited a certain degree of attenuation subsequent to the functional blockade of CD4⁺ TCR-T cells.

6d. In **Figure 5e** (**Figure 5f** of revised Ms), the CD4 T cell controls do not kill as efficiently as the mix of CD4/CD8 T cells without anti-CD4 whereas the CD8 T cells alone do. Please explain.

Response: In **Figure 5e**, the killing ratio is defined as the target cell killing percentage at 12 hours after T cell incubation. Based on the killing curve from **Figure 5c**, CD4⁺ TCR-T cells exhibit a delayed cytotoxic response compared to CD8⁺ TCR-T cells in the initial round of cytotoxicity. It is noteworthy, however, that both cell types demonstrated complete tumor cell elimination within the 48-hour timeframe. Actually, a second round of cytotoxicity assay, akin

to the experiment depicted in **Figure 5c**, also was conducted. In this 2nd challenge experiment, it was observed that the CD4⁺ TCR-T cells in the control group achieved complete eradication of the tumor cells. Conversely, in agreement with the findings in **Figure 5c**, the cytotoxic efficacy of CD8⁺ TCR-T cells had significantly declined, irrespective of the presence of anti-CD4 antibody (**Figure R4, right**). We apologize for the confusion and the related description in the figure legend has been updated.

Figure R4. The second challenge of target tumor cells with the treatment of the anti-CD4 blocking antibody. The 10F04mc transduced CD4⁺, CD8⁺, or CD3⁺ bulk T cells were repeat challenged with HeLa-DR0901 cells in the presence of anti-CD4 blocking antibody or isotype control. The target cell-killing ability was evaluated by the RTCA system. The killing ratio was defined as the target cell killing percentage of each round. Lift: 1st round challenge. Right: 2nd round challenge.

6e. The authors could also look at TCR-modified T cells in vivo and see if CD4 T cells are the ones proliferating when a mix of CD4 and CD8 T cells are injected.

Response: We have indeed conducted an analysis of the proportions of CD4⁺ TCR-T and CD8⁺ TCR-T cells before and after treatment with 10F04mc TCR-T in a mouse xenograft model. The results revealed a significant increase in the proportion of CD4⁺ TCR-T cells 41 days post-infusion. Conversely, the proportion of CD8⁺ TCR-T cells showed a significant decrease (**Figure 3f**).

Figure 3f. Detection of CD4⁺ and CD8⁺ TCR-T cells before and after infusion. NOG mice were subcutaneously injected with 4×10^6 HeLa-DR0901-HPV18E7 cells per mouse. Six days after tumor cell injection, tumor-bearing mice were treated with 5×10^7 10F04mc TCR-T cells by intravenous injection. Mice were euthanized 41 days after TCR-T cell infusion. Murine splenocytes and original TCR-T cells were analyzed by flow cytometry.

It should be discussed in the manuscript why this difference in efficacy between CD4 and CD8 T cells occurs so quickly both in vitro and in vivo. Were any phenotypic markers looked at after rechallenge? CD4 T cells can definitely be efficient cytotoxic cells, but this difference is very striking between the two. What happens to the CD8 T cells? This should be investigated further by flow cytometry phenotyping and functional assays.

Response: Thanks for the question. We think the question is similar with the comment made by the reviewer #1. Therefore, we kindly ask you to refer to our previous response to the 1st major comment of reviewer #1.

7. The methods section is lacking a lot of details (some mentioned above). For flow cytometry, please provide more details on kits and antibodies. Clones should be added (consider a table), and it says anti-mouse TCR antibodies were used, please correct. Please provide more details on antibodies for ELISAs. For in vivo assays, T cell dose is missing. What were humane endpoints in the in vivo studies, i.e. when were mice sacrificed? Indications for statistical testing are missing in several figures.

Response: We thank the reviewer for the suggestion. The revised manuscript has been enriched with more detailed methodologies and reagent information in response to your valuable reminder.

Minor

Cite more of the original papers when referring to specific clinical studies of TCR treatment from review PMID: 36791198 of five clinical trials for NY-ESO-1 TCR, and consider including a recent study: PMID: 37586317.

Response: We thank the reviewer for the suggestion, and we have incorporated these publications in our revised manuscript.

Please change the word pluripotency with multifunctional for T cells secreting multiple cytokines. Pluripotency refers to the capacity of individual cells to initiate all lineages.

Response: We thank the reviewer for the suggestion, and the reviewer's point is well taken.

In Figure 1a, correct survive, should be survival

Check sentences and typos, e.g. line 41: "...is the lacking pre-existing T cells" should be "is the lack of...", line 74: long-term surviving patients rather than long-term survived, line 82: well-immune-response should be corrected,

Response: We appreciate the reviewer for pointing out those grammatical errors. The entire manuscript has been meticulously revised.

Reference:

1. Krishna S, *et al.* Stem-like CD8 T cells mediate response of adoptive cell immunotherapy against human cancer. *Science* **370**, 1328-1334 (2020).
2. Bossio SN, *et al.* CD39+ conventional CD4+ T cells with exhaustion traits and cytotoxic potential infiltrate tumors and expand upon CTLA-4 blockade. *Onc Immunology* **12**, (2023).
3. Bruni L, Diaz M, Castellsagué X, Ferrer E, Bosch FX, de Sanjosé S. Cervical Human Papillomavirus Prevalence in 5 Continents: Meta - Analysis of 1 Million Women with Normal Cytological Findings. *The Journal of Infectious Diseases* **202**, 1789-1799 (2010).

4. Kaminski L-C, *et al.* Cytotoxic T Cell-Derived Granzyme B Is Increased in Severe Plasmodium Falciparum Malaria. *Frontiers in Immunology* **10**, (2019).
5. Lin L, Couturier J, Yu X, Medina MA, Kozinetz CA, Lewis DE. Granzyme B secretion by human memory CD4 T cells is less strictly regulated compared to memory CD8 T cells. *BMC Immunology* **15**, (2014).

REVIEWER COMMENTS

Reviewer #1 (Remarks to the Author):

Manuscript Nr: NCOMMS-23-32950A

Long et al., "An HLA-class II restricted HPV18 E7 specific TCR cloned from a long-term surviving cervical cancer patient induces tumor remission in murine model"

The authors isolated a T cell receptor (TCR) from a patient with human papilloma virus 18 (HPV18) associated cervical carcinoma who stabilized disease for prolonged time periods after multiple antigen stimulating cellular therapy (MASCT). The TCR recognizes a peptide from the E7 HPV18 oncoprotein (aa88-98) presented on the MHC class II molecule HLA-DRA/DRB1*09:01. TCR transduced T cells, in particular those for which the TCR has been converted to a hybrid human-mouse molecule to avoid mispairing with the endogenous human TCR, recognize endogenously HPV18 E7 protein presented on HeLa cells that express HLA-DRA/DRB1*09:01 after transfection. When these cervical carcinoma cells were subcutaneously implanted into immune compromised mice adoptive transfer of TCR transgenic T cells reduced tumor growth with TCR transgenic T cell homing to the tumor microenvironment. Tumor control was better with TCR transgenic CD4+ than CD8+ T cells. An alanine scan of the cognate T cell epitope was performed to identify the important amino acids. Peptides predicted as cognate epitopes based on this motif were not efficiently recognized. Therefore, the authors argued that TCR transfer would not cause immunopathology. The TCR conferred both cytokine production and cytotoxicity to CD4+ and CD8+ T cells but in repeat stimulations TCR transgenic CD4+ T cells also performed better. MHC class II restricted recognition could also be improved by IFN-gamma. The authors were further able to demonstrate long-term persistence of this TCR in the original cancer patient. From these data the authors conclude that their isolated TCR might at least in part be responsible for the long disease stabilization in the patient from which they isolated it and should be explored for treatment of other HLA-DRA/DRB1*09:01 patients with HPV18 positive tumors.

During their manuscript revision the authors have addressed all of my concerns. In order to mimic continuous antigen recognition in the tumor microenvironment the authors report functional and phenotypic changes on TCR transgenic CD4+ and CD8+ T cells. They report interesting differences for CD27, CD39 and PD1. Furthermore, they characterized recognition of E7 sequences of other HPV strains. Finally, they provide more data on the other TCRs that they isolated. Also my minor concerns (HLA-DRB1*09:01 frequency, MASCT description and amount of TCR transgenic T cells that were transferred) were all addressed. Therefore, the manuscript is significantly improved.

However, the two figures for the reviewer R1 and R2 should be incorporated into the supplemental materials.

Reviewer #2 (Remarks to the Author):

The authors are commended for the additional experiments and efforts to satisfy the reviewer comments.

Supplementary figure 4 i) says flow cytometry analysis of HLA-DR*09:01 expression.

Please correct typo in cytometry (cemetery), and assuming that the antibody recognises only HLA-DR, please correct text although it is appreciated that HLA-DR*09:01 is overexpressed.

The reviewer also appreciates the results integrated in Figure 3f showing the % TCR-T cells pre- and post-infusion for CD4+ and CD8+ T cells respectively. Were different donors used for the experiments? In figure 5, it is mentioned that around 70% of the T cells are CD8+ T cells. The transduction efficiency of the two subsets is presented as relatively equal in Figure 3a. However, pre-infusion in Figure 3f, only around 35% of CD8 T cells seemed to be transduced prior to infusion, and very few TCR-T are present in the CD8+ population post-infusion. Was there a general lack of CD8+ T cells, or only TCR-modified CD8+ T cells in the spleen?

In the response to reviewer 1, the authors show that 75-100% of the T cells expressed TIM-3 before the first in vitro tumour challenge. This seems like a very high starting point, and the expression is then reduced after several challenges. Why is this?

POINT-BY-POINT RESPONSE TO REVIEWERS' COMMENTS

REVIEWER COMMENTS

Reviewer #1 (Remarks to the Author):

The authors isolated a T cell receptor (TCR) from a patient with human papilloma virus 18 (HPV18) associated cervical carcinoma who stabilized disease for prolonged time periods after multiple antigen stimulating cellular therapy (MASCT). The TCR recognizes a peptide from the E7 HPV18 oncoprotein (aa88-98) presented on the MHC class II molecule HLA-DRA/DRB1*09:01. TCR transduced T cells, in particular those for which the TCR has been converted to a hybrid human-mouse molecule to avoid mispairing with the endogenous human TCR, recognize endogenously HPV18 E7 protein presented on HeLa cells that express HLA-DRA/DRB1*09:01 after transfection. When these cervical carcinoma cells were subcutaneously implanted into immune compromised mice adoptive transfer of TCR transgenic T cells reduced tumor growth with TCR transgenic T cell homing to the tumor microenvironment. Tumor control was better with TCR transgenic CD4⁺ than CD8⁺ T cells. An alanine scan of the cognate T cell epitope was performed to identify the important amino acids. Peptides predicted as cognate epitopes based on this motif were not efficiently recognized. Therefore, the authors argued that TCR transfer would not cause immunopathology. The TCR conferred both cytokine production and cytotoxicity to CD4⁺ and CD8⁺ T cells but in repeat stimulations TCR transgenic CD4⁺ T cells also performed better. MHC class II restricted recognition could also be improved by IFN-gamma. The authors were further able to demonstrate long-term persistence of this TCR in the original cancer patient. From these data the authors conclude that their isolated TCR might at least in part be responsible for the long disease stabilization in the patient from which they isolated it and should be explored for treatment of other HLA-DRA/DRB1*09:01 patients with HPV18 positive tumors.

During their manuscript revision the authors have addressed all of my concerns. In order to mimic continuous antigen recognition in the tumor microenvironment the authors report functional and phenotypic changes on TCR transgenic CD4⁺ and CD8⁺ T cells. They report interesting differences for CD27, CD39 and PD1. Furthermore, they characterized recognition

of E7 sequences of other HPV strains. Finally, they provide more data on the other TCRs that they isolated. Also my minor concerns (HLA-DRB1*09:01 frequency, MASCT description and amount of TCR transgenic T cells that were transferred) were all addressed. Therefore, the manuscript is significantly improved.

However, the two figures for the reviewer R1 and R2 should be incorporated into the supplemental materials.

Response: The authors greatly appreciate your helpful and valuable comments and I would like to thank you again for your precious time and effort in reviewing the revised manuscript. The two figures for the reviewers R1 and R2 have been incorporated into the supplemental materials as suggested.

Reviewer #2 (Remarks to the Author):

The authors are commended for the additional experiments and efforts to satisfy the reviewer comments.

Supplementary figure 4 i) says flow cytometry analysis of HLA-DR*09:01 expression.

Please correct typo in cytometry (cemetery), and assuming that the antibody recognises only HLA-DR, please correct text although it is appreciated that HLA-DR*09:01 is overexpressed.

Response: We apologize for the writing error and the description in the figure legend has been corrected.

The reviewer also appreciates the results integrated in Figure 3f showing the % TCR-T cells pre- and post-infusion for CD4⁺ and CD8⁺ T cells respectively. Were different donors used for the experiments?

Response: The results presented in Figure 3f represent findings from two biologically independent experiments with consistent observations. In total, three different healthy donor PBMC-derived TCR-T cells were analyzed before infusion and after being infused into HPV18E7⁺HLA-DR*09:01⁺ tumor-bearing NOG mice. Consistent result was observed among all three tested donors, showing a marked increase in the proportion of CD4⁺ T cells and a corresponding decrease in the proportion of CD8⁺ T cells within TCR-T cells following infusion

(Figure R5).

Figure R5. Changes in the proportions of CD4⁺ and CD8⁺ T cells in TCR-T cells before and after infusion. NOG mice were injected subcutaneously with 4×10^6 HeLa-DR0901-HPV18E7 cells per mouse and treated with 5×10^7 10F04mc transduced TCR-T cells 6 days later. Splenocyte TCR-T cells (after infusion) and original TCR-T cells (before infusion) were analyzed by flow cytometry at the endpoint of the experiments.

In figure 5, it is mentioned that around 70% of the T cells are CD8⁺ T cells. The transduction efficiency of the two subsets is presented as relatively equal in Figure 3a. However, pre-infusion in Figure 3f, only around 35% of CD8 T cells seemed to be transduced prior to infusion, and very few TCR-T are present in the CD8⁺ population post-infusion. Was there a general lack of CD8⁺ T cells, or only TCR-modified CD8⁺ T cells in the spleen?

Response: Sorry for the confusion. Figure 3f actually shows the dynamic changes in the proportions of CD4⁺ and CD8⁺ T cells within TCR-T cell populations rather than the transduction efficiency. Indeed, as demonstrated in **Figure 3a**, this TCR exhibits similar transduction efficiencies in both subsets. As shown in **Figure R5**, the initial proportions of CD4⁺ and CD8⁺ T cells vary among different donors, but following reinfusion, there is a significant shift in the proportions of both cell types. These results suggest that CD4⁺ TCR-T cells may undergo more robust expansion in tumor-bearing mice.

In the response to reviewer 1, the authors show that 75-100% of the T cells expressed TIM-3 before the first in vitro tumour challenge. This seems like a very high starting point, and the expression is then reduced after several challenges. Why is this?

Response: Thank you for your question. The expression of Tim-3 on the surface of CD4⁺ and

CD8⁺ T cells is significantly upregulated after activation with anti-CD3/CD28 stimulation¹. Our TCR-T cells were utilized in co-culture experiments with tumor cells for approximately 8 days post initial activation, at which point Tim-3 expression levels were found to be comparatively high. Additionally, as previously reported², TIM-3 is primarily expressed in effector T cells (T eff), consistent with the majority of cells being in the Teff phenotype (**Figure 5b**). Exhausted T (Tex) cells exhibit weak tumor-killing capabilities, whereas Teff cells display stronger activities, as observed consistently in the first two rounds of co-culture experiments (**Figure 5c**). Furthermore, in the *in vitro* repeated challenge experiments, we also did not detect the expression of other classical T cell exhaustion markers, such as PD1 (**Supplementary figure 9d**). Therefore, based on those results, we believe that the observed elevated and gradually decreasing expression of Tim3 may be primarily attributed to *in vitro* activation and prolonged culturing period.

Reference

1. Avery L, Filderman J, Szymczak-Workman AL, Kane LP. Tim-3 co-stimulation promotes short-lived effector T cells, restricts memory precursors, and is dispensable for T cell exhaustion. *Proceedings of the National Academy of Sciences* **115**, 2455-2460 (2018).
2. Sabins NC, *et al.* TIM-3 Engagement Promotes Effector Memory T Cell Differentiation of Human Antigen-Specific CD8 T Cells by Activating mTORC1. *The Journal of Immunology* **199**, 4091-4102 (2017).

REVIEWERS' COMMENTS

Reviewer #2 (Remarks to the Author):

The authors have answered all my questions.

POINT-BY-POINT RESPONSE TO REVIEWERS' COMMENTS

REVIEWERS' COMMENTS

Reviewer #2 (Remarks to the Author):

The authors have answered all my questions.

Response: Thank you.